# ZeroC: A Neuro-Symbolic Model for Zero-shot Concept Recognition and Acquisition at Inference Time

**Tailin Wu**
Department of Computer Science
Stanford University
`tailin@cs.stanford.edu`

**Megan Tjandrasuwita**
Department of Computer Science
California Institute of Technology
`megantj@mit.edu`

**Zhengxuan Wu**
Department of Computer Science
Stanford University
`wuzhengx@cs.stanford.edu`

**Xuelin Yang**
Department of Computer Science
Stanford University
`xyang23@cs.stanford.edu`

**Kevin Liu**
Department of Computer Science
Stanford University
`liuk@cs.stanford.edu`

**Rok Sosič**
Department of Computer Science
Stanford University
`rok@cs.stanford.edu`

**Jure Leskovec**
Department of Computer Science
Stanford University
`jure@cs.stanford.edu`

## Abstract

Humans have the remarkable ability to recognize and acquire novel visual concepts in a zero-shot manner. Given a high-level, symbolic description of a novel concept in terms of previously learned visual concepts and their relations, humans can recognize novel concepts without seeing any examples. Moreover, they can acquire new concepts by parsing and communicating symbolic structures using learned visual concepts and relations. Endowing these capabilities in machines is pivotal in improving their generalization capability at inference time. In this work, we introduce Zero-shot Concept Recognition and Acquisition (ZeroC), a neuro-symbolic architecture that can recognize and acquire novel concepts in a zero-shot way. ZeroC represents concepts as graphs of constituent concept models (as nodes) and their relations (as edges). To allow inference time composition, we employ energy-based models (EBMs) to model concepts and relations. We design ZeroC architecture so that it allows a one-to-one mapping between a symbolic graph structure of a concept and its corresponding EBM, which for the first time, allows acquiring new concepts, communicating its graph structure, and applying it to classification and detection tasks (even across domains) at inference time. We introduce algorithms for learning and inference with ZeroC. We evaluate ZeroC on a challenging grid-world dataset which is designed to probe zero-shot concept recognition and acquisition, and demonstrate its capability. [1]

---

[1]Project website and code can be found at `http://snap.stanford.edu/zeroc/`.

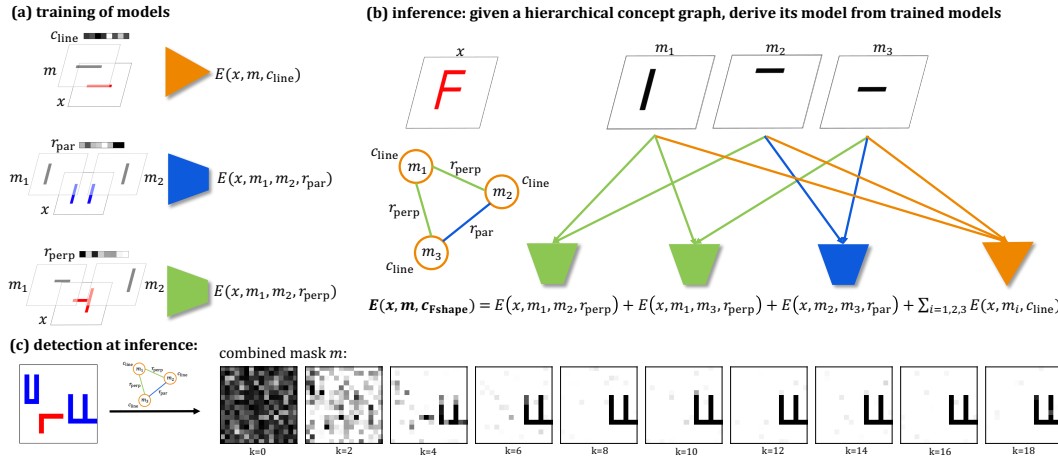

Figure 1: **Composition of Concept Models.** We demonstrate the model composition for a novel, hierarchical concepts on the recognition of letter F. **(a)** During training, we learn the models for constituent concepts, the concept "line" in this case, and relations, which are "parallel" and "perpendicular". **(b)** During inference, we take the concept graph of F and use it to derive the model for F from the models of its constituents. Note that no training is performed on the hierarchical concept F. **(c)** Example of detecting hierarchical concept on pixel level with ZeroC during inference in Sec. 3, using Eq. (1).

# 1 Introduction

Humans learn in diverse ways. Besides learning from demonstrations of a novel concept, humans can also learn concepts on a high-level. Consider learning the "rectangle" concept, for example. Suppose that one has never seen such a concept, but has already mastered the concept of "line" and relations of "parallel" and "perpendicular"; s/he can easily master the "rectangle" concept when told that a "rectangle" consists of two pairs of "lines", the lines *within* the pairs are "parallel," and the lines *between* the pairs are "perpendicular". Then s/he can directly use this newly mastered concept to recognize "rectangles" in novel images.

Such zero-shot concept *recognition* capability is still beyond the reach of deep learning models, which require many examples (as in typical supervised learning), or many tasks drawn from the same distribution (as in few-shot learning) to learn a novel concept. Moreover, along with the above symbolic to instance direction, humans can easily do the *reverse*. Suppose again that we haven't seen the "rectangle" concept, but have already mastered concepts of "line" and relations of "parallel" and "perpendicular". When seeing a novel image which contains an instance of a rectangle, humans can easily decompose it into its constituent concepts of "lines" and their relational graph structure (from instance to symbolic). Moreover, humans can then communicate with each other about this new concept, allowing the transfer of knowledge in a high-level and in a succinct way. Such zero-shot concept *acquisition* capability at inference time is beyond the reach of today's machine learning and AI systems. Above all, endowing the above two capabilities of zero-shot concept recognition and acquisition capability to machines at inference time, will allow them to tackle more diverse tasks.

Past efforts have attempted to address aspects of the above problem of recognizing and acquiring novel concepts at inference time in a zero-shot manner. An essential component is compositionality, *i.e.* the ability to compose new concepts from elementary ones. [1] and [2] introduce techniques for compositional generation in the context of energy-based models (EBMs) and variational autoencoders (VAEs), respectively. However, these works only consider composing factors of variation (*e.g.*, color, position, smiling, young). In order to recognize concepts that consist of constituent concepts as parts, the structure of objects and their relationships are equally important. For instance, given 4 "line" concepts, without specifying their relational graph structure, they can easily form a trapezoid or a generic parallelogram. [3] designed a transformer-based relational architecture for learning explicit relations, and showed that it can generalize to new objects for the relation. However, this architecture cannot generalize to concepts with more complex relational graph structure.

It remains an open problem of how to design an architecture that has the capability to compose novel concepts at inference time, based on its internal relational graph structure of prior learned concepts

and relations. A separate line of works [4, 5, 6, 7, 8] tackle zero-shot classification, but are limited to novel composition of label features for concepts, and ignore relation features as well as the graph structure formed by concepts and relations.

Here we introduce our ZeroC architecture, which is a model designed towards zero-shot concept recognition and acquisition. It models a visual concept or relation as an Energy-Based Model (EBM) with 3 inputs: an image, a mask (or two masks for relation) denoting a set of pixels within the image, and a concept/relation label string. It returns a low scalar energy value if the mask(s) correctly indicates the concept or relation within the image, and a high energy value otherwise. After training, the concept and relation EBMs are able to perform classification of concept/relation labels given an image, and detection of concept/relation mask(s) given an image and concept/relation label. The *core* contribution of this work is ZeroC's architecture and inference algorithms that allow a one-to-one correspondence between a specification of a relational graph structure of a novel concept and its corresponding composed concept model that can also perform similar downstream classification and detection, which enables zero-shot concept recognition and acquisition at inference time.

To perform zero-shot concept recognition at inference time, ZeroC can simply compose a new concept EBM, by summing over the constituent concept and relation EBMs according to the specified graph structure. This new concept EBM has the same input-output format as its constituent concept EBMs, allowing the same capability of classification and detection. Note that through the above process, ZeroC have mastered a concept without seeing an image, only symbolic instruction, thus it is called "zero-shot". To perform zero-shot concept acquisition, we introduce an algorithm that given an instance (image) of a novel concept, parse it into a relational graph of prior learned concepts and relations. Such relational graph can then be transferred, even cross-domain, allowing models independently trained in a different domain to be able to zero-shot classify and detect this concept in its own domain.

There are currently no specialized benchmarks to test such zero-shot concept recognition and acquisition capabilities. Inspired by the Abstraction and Reasoning Corpus (ARC) [9], we have created a synthetic, grid-world based dataset with tasks that capture the essence of the above capabilities that are deceptively simple for humans, but very hard for neural models. We demonstrate that our model, trained to classify and detect elementary visual concepts and relations, is able to classify and detect novel concepts at inference time, being given a zero-shot symbolic graph structure. We also compare with a state-of-the-art zero-shot learning model CADA-VAE [8]. Due to that it is not designed for such graph-based composition, it significantly underperforms our ZeroC. Furthermore, we show that two independently trained models, one on 2D images and the other on 3D images, are able to acquire hierarchical concepts from each other by communicating the graph structure and then perform classification and detection of these hierarchical concepts in their respective domains.

## 2 Method

In this section, we describe the ZeroC architecture for recognizing and acquiring novel visual concepts at inference time[2]. We give an overview of our approach, describe how it performs classification and detection, how it supports zero-shot concept recognition, how independently-trained models can acquire hierarchical concepts from each other and, finally, introduce its learning method.

### 2.1 An Overview of ZeroC Architecture

The key components of ZeroC are concepts and relations. Each concept consists of a graph and an energy-based model. The concept graph describes the concept as a composition of its constituent concepts and relations between them. The base concepts that do not have any constituent concepts are called elementary concepts. Their graph is a singleton. The concept energy-based model is used to recognize the concept in the input data. Each relation is represented in ZeroC also with a graph and an energy-based model. The relation graph is simply an edge that connects the two related concepts. A hierarchical concept is a concept composed of constituent concepts as nodes and relations as edges according to a graph structure.

---

[2]In this paper, we focus on visual concepts. However, our framework is fully general and may be applied to other input modalities such as graphs, natural language, etc.

During training, energy-based models are learned from images and concept or relation labels. During inference, the learned models are used to recognize concepts seen during training or more complex hierarchical concepts which were not seen before. For concepts seen during training, their learned models are applied. For new hierarchical concepts not seen during training (*i.e.*, without learned model), their models are derived from the graph of the new hierarchical concept and energy-based models of their constituent concepts and relations. Concepts and relations in ZeroC can be viewed as templates for objects and their connections, which then get grounded during inference with a specific image, where those objects and relations are assigned actual values. ZeroC also handles object masks, which indicate object locations in the image.

## 2.2 Zero-shot Concept Recognition and Acquisition

Formally, we model a concept with $E_{X,M,C}(x, m, c)$[3], which maps an image $x \in \mathbb{R}^D$, a mask $m \in [0, 1]^D$ and a concept label[4] $c$ to a scalar energy. Similarly, we model relations with $E_{X,M_1,M_2,R}(x, m_1, m_2, r)$, where $m_1$ and $m_2$ are a pair of masks indicating two objects in the image $x$, and $r$ is a relation label between the objects. The concept and relation models have a probabilistic interpretation. For example, the energy function $E_{X,M,C}(x, m, c)$ corresponds to the joint probability of

$$P_{X,M,C}(x, m, c) = \frac{1}{Z_C}\exp\left(-E_{X,M,C}(x, m, c)\right)$$

where $Z_C = \sum_{c \in C} \int e^{-E_{X,M,C}(x,m,c)} \, dx \, dm$ is a normalizing constant. Therefore, if the mask $m$ is actually masking an object that belongs to concept $c$ in image $x$, then $P_{X,M,C}(x, m, c)$ will be high and $E_{X,M,C}(x, m, c)$ will be low, and vice versa. Essentially, the models define an energy landscape for their respective multi-modal inputs that give low energy if the mask correctly identifies its corresponding concept.

Next, we show how ZeroC performs detection, classification, and models hierarchically composed concepts. We use the term HC-EBM to denote concept models and R-EBM for relation models.

**Detection.** We want to infer the location mask $m$ of the concept $c$, given image $x$. Probabilistically, we are computing $P_{M|X,C}(m|x, c)$. To perform detection, we employ Stochastic Gradient Langevin Dynamics (SGLD) to sample masks on the energy landscape [10]:

$$\tilde{m}^k = \tilde{m}^{k-1} - \frac{\lambda}{2}\nabla_m E_{X,M,C}(x, \tilde{m}^{k-1}, c) + \omega^k, \ \omega^k \sim \mathcal{N}(0, \lambda) \tag{1}$$

where $\tilde{m}^k$ is the inferred mask at the $k$th iteration, $k = 1, 2, ...K$. Applying [11], as $K \to +\infty$ and $\lambda \to 0$, we generate samples from the distribution of $P_{M|X,C}(m|x, c) = \frac{1}{Z_{x,c}}\exp(-E_{X,M,C}(x, m, c))$, where $Z_{x,c} = \int \exp(-E_{X,M,C}(x, m, c)) \, dm$ is a normalizing constant. In practice, we use a finite $K$ to generate samples[5] $\tilde{m}_n^K, n = 1, 2, ...N$ given $x, c$.

**Classification.** We want to determine whether the concept $c$ appears in a given image $x$, *i.e.* compute $P_{C|X}(c|x)$. We need to marginalize over the mask $m$:

$$P_{C|X}(c|x) = \frac{P_{X,C}(x, c)}{P_X(x)} = \frac{\int P_{X,M,C}(x, m, c) \, dm}{\sum_{c \in C} \int P_{X,M,C}(x, m, c) \, dm}$$
$$= \frac{\int \exp\left(-E_{X,M,C}(x, m, c)\right) \, dm}{\sum_{c \in C} \int \exp\left(-E_{X,M,C}(x, m, c)\right) \, dm}$$

We again use SGLD [10] in Eq. (1) to generate $N$ samples $\tilde{m}_n^K, n \in [N] = \{1, 2, ...N\}$ given $x, c$, and approximate the above integral using maximum a posteriori (MAP) estimation:

$$P_{C|X}(c|x) \simeq \frac{\max_{n \in [N]} \exp\left(-E_{X,M,C}(x, \tilde{m}_n^K, c)\right)}{\sum_{c \in C} \max_{n \in [N]} \exp\left(-E_{X,M,C}(x, \tilde{m}_n^K, c)\right)} \tag{2}$$

---

[3]We will use capital letters (*e.g.* $X, M$) to represent random variables, and small ones (*e.g.* $x, m$) to represent their instances.

[4]The concept label $c$ is a categorical variable, that we use to refer to a concept, like "line", "rectangle", or "cat". In the following, if without confusion, we will refer to a concept using its label $c$.

[5]In this paper, we use $i$ to index different concepts, $j$ to index relations, and $n$ to index example images.

In practice, we only need to find the concept with the highest value in the numerator.

**Zero-shot recognition of novel concepts.** To master a novel hierarchical concept and directly use it for classification and detection only given its relational graph structure, we need a way to compose the previous concept and relation energy based models. Here we introduce the hierarchical composition rule, using an English letter "F" as an illustrative example[6].

The concept F has one constituent concept, a "line", and two relations, "parallel" and "perpendicular". The models for constituents are learned during training. During inference, these models are combined, using the concept graph, into a combined energy model for the letter F. Essentially, the models of all the recognized constituent concepts and relations are added together. Note that although F contains three lines, only one "line" model is needed. The concept graph acts as a template and recognized line instances (objects) are matched with nodes in the template to obtain actual models. The "line" model is instantiated three times with specific values for the three identified lines and the three models plus their corresponding relations models are used to derive the hierarchical model. In addition to the models, we also need to combine the masks of all the recognized constituent objects.

Formally, we define the following composition rule for HC-EBMs plus the masks to be combined.

**Definition 2.1. Hierarchical Composition Rule:** Let a hierarchical concept $c$ have graph $\mathcal{G} = (\mathcal{C}, \mathcal{R})$, where $\mathcal{C}$ are constituent concept nodes and $\mathcal{R}$ are relation edges. During inference, these nodes and edges are matched to recognized concept objects and their relationships, which provides their models and masks. The combined model $E_{X,M,C}(x, m, c)$ is then a sum of the models for all the nodes and edges in the graph and the mask $M$ is the maximum of all the masks:

$$E_{X,M,C}(x, m, c) = \sum_{c_i \in \mathcal{C}} E_{X,M_i,C}(x, m_i, c_i) + \sum_{r_j \in \mathcal{R}} E_{X,M_{j1},M_{j2},R}(x, m_{j1}, m_{j2}, r_j) \quad (3)$$

$$M := \max\{\{M_i\}, \{M_{j1}\}, \{M_{j2}\}\}$$

## 2.3 Zero-shot Concept Acquisition at Inference Time

Here, we introduce techniques for acquiring novel concept graph, which is useful for sharing high-level knowledge transfer between independently trained models. Such capability may have implications in future scenarios where a hypothetical self-driving car communicating the structure of a novel road sign to other cars on the road, or a 2D vision model learning a novel object and transferring it to a more accurate 3D depth model for inference. To achieve this, we need to establish an equivalence relation between concepts in different domains (*e.g.* in 2D and 3D images).

**Definition 2.2. Structural Equivalence for HC-EBM:** Two HC-EBMs are structurally equivalent, if their graphs are isomorphic, and their constituent HC-EBMs are recursively structurally equivalent.

For example, HC-EBMs for a "rectangle" concept in 3D and 2D images are structurally equivalent if they have the same decomposition into two "parallel-line" HC-EBMs connected by a "connect" R-EBM, and the "parallel-lines" have the same decomposition into two "line" HC-EBMs connected by a "parallel" R-EBM. Even though the HC-EBMs are grounded in different domains, they have the same abstract structure, so they represent the same hierarchical concept.

With structural equivalence, independently-trained HC-EBMs from different domains can acquire hierarchical concepts via the graph structure. The algorithm works as follows (see Appendix A.5): the image is parsed to decompose it into a graph of concepts and relations previously learned by HC-EBM and R-EBM, then the graph is used in a different domain to compose its HC-EBM for the new concept. The key step of this process is the first, parsing step.

The parsing step can be seen as an inverse of the Hierarchical Composition Rule and is critical in allowing the model to recognize novel hierarchical concepts in its domain using prior-learned concepts and relations from other domains, thus facilitating downstream knowledge transfer between domains. Alg. 1 provides further details. Steps 1-2 infer the concept instances in image $x$ using SGLD (Eq. 1). Here the energy $E^{(p)}(x, \mathcal{M}_C, c)$ is the summation of HC-EBMs on independent masks, each mask $m_{i_l}$ representing a potential concept instance belonging to concept $c_i$. $E^{(\text{overlap})} = \max\left(0, \sum_{m_{i_n} \in \mathcal{M}_C} m_{i_n} - 1\right)$ penalizes overlapping masks. Since the image probably contains fewer concept instances than given, some masks are empty with all-zero values; Step 3 removes these.

---

[6] For clarity, the concept graph shown has been simplified from the real model used in the experiments.

**Algorithm 1 Parsing Hierarchical Concept From Image**

---

**Require:** HC-EBM $E^{(C)}$ with prior-learned concepts $\{c_1, c_2, ...c_I\}$, R-EBM model $E^{(R)}$ with prior-learned relations $\{r_1, r_2, ...r_J\}$.
**Require:** Image $x$, containing unseen hierarchical concept $c$.
**Require:** Maximum number of instances $N_i$ for each concept $c_i$, $i = 1, 2, ...I$.
1: $E^{(p)}(x, \mathcal{M}_C, c) := \sum_{i=1}^{I} \sum_{n=1}^{N_i} E^{(C)}(x, m_{i_n}, c_i)$
$\quad + \lambda_1 E^{(\text{overlap})}(\mathcal{M}_C)$, where $\mathcal{M}_C := \{m_{i_n}\}, i_n =$
$\quad 1, ...N_i$ for $i = 1, ...I$, is the set of all masks.
2: $\mathcal{M}_C \leftarrow \textbf{SGLD}_{\mathcal{M}_C}(E^{(p)}(x, \mathcal{M}_C, c))$    *// Using Eq. 1*
3: $\mathcal{M}_C \leftarrow$ Remove-empty-masks$(\mathcal{M}_C)$
4: $\mathcal{R} \leftarrow \{r_{j_1 j_2} | \textbf{Classify}(E^{(R)}; x, m_{j_1}, m_{j_2}), \forall(m_{j_1}, m_{j_2}) \in \mathcal{M}_C\}$    *// Using Eq. 2 to classify*
5: $\mathcal{G} \leftarrow$ Build-Graph$(\mathcal{M}_C, \mathcal{R})$
6: **return** $\mathcal{G}$

---

Step 4 classifies relations (using Eq. 2) between pairs of detected concept instances. Step 5 combines the detected concept instances and their relations to build the graph $\mathcal{G}$ for this new hierarchical concept $c$.

## 2.4 Learning

Since a standard EBM-training method yields poor performance as shown later in our experiments, we describe here our approach to train the models. Further details of our training algorithm are given in Alg. 2 in Appendix A.1. We are optimizing the following objective:

$$L = \frac{1}{N} \sum_{n=1}^{N} \left( L_n^{(\text{Improved})} + \alpha_{\text{pos-std}} L_n^{(\text{pos-std})} + \alpha_{\text{em}} L_n^{(\text{em})} + \alpha_{\text{neg}} L_n^{(\text{neg})} \right) \tag{4}$$

The expression of each term is given in Appendix A.2. The first loss $L^{(\text{Improved})}$ is the objective proposed in [12], a state-of-the-art EBM training technique. However, to enable the challenging zero-shot concept recognition and acquisition, we need more suitable inductive biases. The next three terms inject the right inductive bias for the task. $L^{(\text{pos-std})}$ makes sure that the positive energy have similar level, so the composed concept EBM can identify all constituent concepts and relations, without one constituent EBM having too low energy and only recognizing it. The empty-mask regularization $L^{(\text{em})}$ makes sure that the when the mask is empty, its energy is between the positive and negative energy, which we prove in Appendix A.3 that it is the necessary condition for correctly discover the underlying graph in Sec. 2.3. The $L^{(\text{neg})}$ additionally provide additional negative examples, encouraging discovering full concept instead of a part of it (see Appendix A.2 for details).

# 3 Experiments

In this section, we set out to answer the following two questions: (a) given a specification of graph structure for a novel hierarchical concept, can the composed concept model successfully perform classification and detection tasks? (b) Can HC-EBM acquire a novel concept from HC-EBM trained from another domain?

We also evaluated our model in a challenging setting by comparing with other existing approaches in a controlled and systematic way. Since there are no suitable dataset, we designed a Hierarchical-Concept corpus, a dataset based on grid-world images. The dataset comes with testing images for the classification and detection tasks which contain more complex, hierarchical concepts, composed from simpler concepts in the training set (See Appendix A.9 for details). The concept instances on our images have varying locations, size, and relative positions, making them more difficult than fixed-location fixed-size inference tasks such as in [3]. We also provide 2D and 3D versions of images for demonstrating concept acquisition across domains.

Table 1: Performance of models on classification accuracy for hierarchical dataset 1 and 2 (%) and on detection for hierarchical concepts with distractors. For the latter task, we use the pixel-wise intersection-over-union (IoU) (%) as our metrics. The bold fonts in the tables indicate the best model comparing with baselines. The "Statistics" in classification predicts the class that has the most global label fraction.

| Model | Classification (acc.) | | Detection (IoU) | |
|---|---|---|---|---|
| | HD-Letter | HD-Concept | HD-Letter+distractor | HD-Concept+distractor |
| Statistics | 46.5 | 53.5 | 5.69 | 12.6 |
| Heuristics | (–) | (–) | 42.3 | 29.2 |
| CADA-VAE [8] | 18.0 | 66.0 | (–) | (–) |
| **ZeroC (ours)** | **84.5** | **70.5** | **72.5** | **84.7** |
| ZeroC composition without R-EBM | 62.5 | 32.5 | 45.3 | 84.3 |
| ZeroC composition without HC-EBM | 67.0 | 55.0 | 67.7 | 78.4 |
| ZeroC without $L^{(pos\text{-}std)}$ | 43.6 | 65.5 | 76.1 | 81.5 |
| ZeroC without $L^{(neg)}$ | 64.5 | 59.0 | 60.0 | 84.2 |
| ZeroC without $L^{(em)}$ | 81.5 | 61.0 | 68.0 | 86.0 |
| ZeroC with only $L^{(Improved)}$ | 27.5 | 55.5 | 49.1 | 81.7 |

## 3.1 Zero-shot Classification and Detection of Novel Concepts

To test recognition of novel concepts, we designed two datasets consisting of different concept and relation types. The HD-Letter hierarchical dataset consists of concept instances of "line" and relation instances of "parallel", "perp-edge" (perpendicular and touching edge), "perp-mid" (perpendicular and touching middle), together with distractor objects. Examples are provided in the form of 3-tuples $(image, mask, concept)$ for concepts, and $(image, mask_1, mask_2, relation)$ for relations. ZeroC models are trained via objective Eq. 4 (See Appendix A.9 for examples of training datasets). At inference time, the models need to perform classification and detection on novel images with more complicated English characters of "E-shape", "F-shape" and "A-shape", given their structure graphs with up to 4 nodes and 6 edges. For detection, the images also contain a few distractors (concepts unrelated to the ones the model is predicting). The model is asked to return a mask indicating which pixels belong to an instance of the specified novel concept. Although looking simple, this is actually a very challenging task. because in order to solve the problem of classifying/detecting hierarchical concepts, the model needs to solve a subgraph isomorphism problem, which is NP-hard. For example, take the problem of detecting "E" shape in an image with distractor of "T" and "Rectangle" (Fig. 2 (a) first subfigure), a graph with 10 nodes (lines), 45 possible edges. Note that during training time, a model has only learned "line" concept, and relations "parallel", "perp-edge" (perpendicular, touching edge), and "perp-mid", but not the overall "E" shape. At inference time and given the "E" concept graph with 4 nodes (lines), 6 edges (relations), the model needs to find the "E" concept subgraph within the large image graph. This involves $C_{10}^4 \times 4! = 5040$ possible mask assignments. Additionally, a model may not perfectly detect the masks for low-level concepts.

For more complex concepts and relations, we designed the HD-Concept hierarchical dataset, which consists of training concepts of "E-shape" and "rectangle", and training relations of "inside", "enclose" and "non-overlap". The hierarchical concepts to be classified and detected are three characters which we term Concept1, Concept2 and Concept3, as the ground-truth masks in Fig. 2 (a) indicate. The three hierarchical concepts have the same multiset of concepts, but a different relation structures.

As the closest setting to our classification task is zero-shot classification, we compare ZeroC to a state-of-the-art zero-shot learning algorithm CADA-VAE [8], which we adapted to our setting, using the set of concepts and relations as feature embeddings to represent the concept graph (see Appendix A.6 for details). Additionally, we evaluated a "Statistics" baseline which samples random pixels on the image based on the global statistics of pixel occurrence, and a "Heuristics" baseline that randomly chooses a same-color connected object. We use a CNN-based architecture for ZeroC, as given in Appendix A.4. We add comparisons with ablation of different aspects of ZeroC, including without HC-EBM terms or R-EBM terms in the composition rule of Def. 2.1 (Eq. 3), and without different regularization terms of the objective in Eq. 4.

Fig. 2 shows a demonstration of this experiment. From performance results in Table 1, we see that ZeroC achieves classification accuracy of 84.5% and 70.5% on HD-Letter and HD-Concept, respectively, both higher than CADA-VAE. The gap is more significant in HD-Letter, where there is a larger distribution shift for hierarchical concepts (as can be seen in Appendix A.9), which the

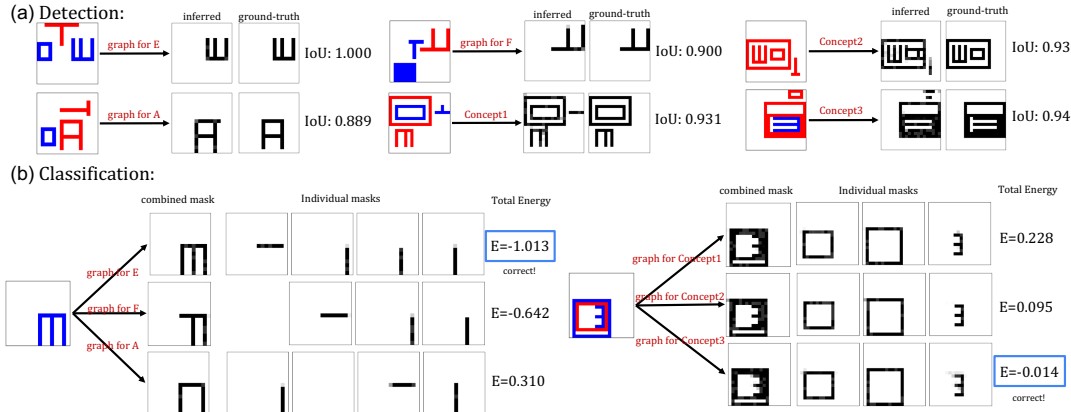

Figure 2: **Sample of ZeroC's Inference Results.** Tasks are (a) pixel-wise detection, and (b) classification, for the datasets of HD-Letter (consisting of concept E-shape, F-shape and A-shape) and HD-Concept Dataset (consisting of Concept1, Concept2, Concept3). We see that ZeroC has high pixel-wise detection IoU for the specified concepts. For classification, ZeroC uses each candidate concept's graph and performs SGLD on the composed HC-EBM to infer the mask and compute its energy. We see that ZeroC gives lower energy to the correct concept, even when sometimes candidate concepts have similar inferred mask ((b) right).

Table 2: Performance of models on acquiring concepts between models and domains at inference time (%).

| Model | Domain 1 (2D image) Parsing | | Domain 2 (3D image) | |
|---|---|---|---|---|
| | Isomorphism (acc.) ↑ | Edit distance ↓ | Classification (acc.) ↑ | Detection (IoU) ↑ |
| Statistics | 2.33 | 3.14 | 33.3 | 2.53 |
| Mask R-CNN [13]+relation classification | 35.5 | 1.01 | (–) | (–) |
| **ZeroC$_1$ → ZeroC$_2$ (ours)** | **72.7** | **0.50** | **60.7** | **94.4** |
| ZeroC$_1$ without $L^{(pos\text{-}std)}$ → ZeroC$_2$ | 55.2 | 1.57 | 54.7 | 90.5 |
| ZeroC$_1$ without $L^{(neg)}$ → ZeroC$_2$ | 53.5 | 0.99 | 52.5 | 92.1 |
| ZeroC$_1$ without $L^{(em)}$ → ZeroC$_2$ | 50.7 | 1.58 | 51.3 | 95.4 |
| ZeroC$_1$ with only $L^{(Improved)}$ → ZeroC$_2$ | 11.5 | 2.00 | 50.5 | 94.7 |
| ZeroC$_2$ with ground-truth graph (upper bound) | (–) | (–) | 61.8 | 94.2 |

joint embedding of the image and features learned by CADA-VAE is insufficient to handle. The reason for the low accuracy is that CADA-VAE is not able to address these out-of-domain distribution shifts (See Appendix A.7 for details). During inference, its embedding (multi-hot vector) for the graph structure can contain up to 10 hots (4 lines, 6 edges), while during training, it is only up to 1-hot. This example also shows the intrinsic difficulty of the task. In addition, Table 1 and Fig. 2 (a) show that ZeroC is able to detect the hierarchical concepts in the presence of distractors, and that its performance is significantly better than "Statistics" and "Heuristics" baselines.

The results of our ablation studies in Table 1 show that both the HC-EBM and R-EBM terms in Eq. 3 are key to successful classification and detection. This is likely because without relations and using only concept terms, ZeroC may lose its ability to distinguish between different hierarchical concepts; for example, both "E-shape" and "A-shape" have 4 lines but different relation structures. Moreover, we see that both $L^{(pos\text{-}std)}$ and $L^{(neg)}$ improve classification and detection performance.

### 3.2 Acquiring Novel Hierarchical Concepts Across Domains

We show that ZeroC can acquire novel hierarchical concepts across models and domains (Fig. 3). We train a ZeroC model in domain ZeroC$_1$, where the images are 2D and one-hot color-coded, containing the the same training concepts and relation types as in HD-Letter in Sec. 3.1. *Independently*, we train another model in a different domain ZeroC$_2$, where the images contain the same set of concepts and relations, but are viewed in 3D from a certain camera angle, have larger size, and use RGB colors. At inference time, each test task consists of a tuple of three images showing the hierarchical concepts in the first domain to ZeroC$_1$. ZeroC$_1$ is only allowed to send symbolic information, up to a few bits, to ZeroC$_2$. Then ZeroC$_2$ performs classification and detection tasks on three example images in the second domain. In addition, we also evaluate whether ZeroC$_1$ can parse the concept graph of the hierarchical concept correctly, using the metric of graph edit distance $d_{edit}$ and graph-isomorphism accuracy w.r.t. the ground-truth. Note that the graph-isomorphism accuracy is a stringent metric,

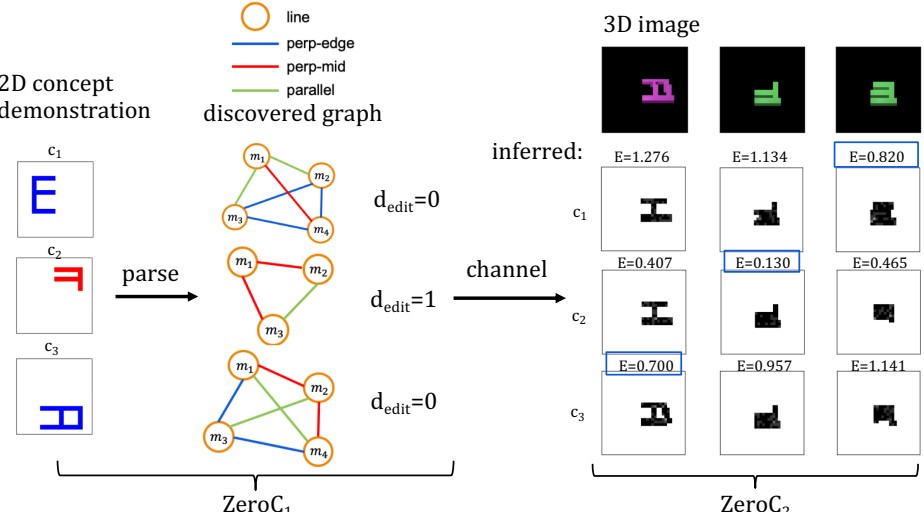

Figure 3: **Acquiring Hierarchical Concept between Domains.** The figure shows actual example tasks and results. At inference, $ZeroC_1$ sees the 2D demonstration of three images showing three unseen concepts $c_1$, $c_2$, $c_3$. It first parses each image into respective concept graphs. We see that except for $c_2$ that has an edit distance $d_{edit}$ of 1, the others have perfect parsing. $ZeroC_1$ then sends to $ZeroC_2$ the parsed concept graphs, which $ZeroC_2$ uses to perform classification that selects which 3D image corresponds to each concept.

which is only 1 if the parsed graph is isomorphic to the ground-truth, and 0 otherwise[7]. We compare with a strong baseline of Mask R-CNN [13] for object detection together with relation classification. The relation classification network is a CNN which takes as input a pair of embeddings produced by the Mask R-CNN, and predicts the label of the relation (see Appendix A.8 for details). We did extensive tuning of the Mask R-CNN approach to obtain its best performance. We also evaluate the performance of $ZeroC_2$ given the ground-truth structure graph, as an ideal scenario of perfect parsing, providing an upper bound for the performance.

Table 2 shows that $ZeroC_1$ achieves 72.7% graph isomorphism accuracy for parsing the hierarchical concepts in the first domain, significantly higher than Mask R-CNN + relation classification. After sending the concept graphs in the second domain, $ZeroC_2$ achieves a classification accuracy of 60.7% and detection IoU of 94.4%. Without this information and relying on global statistics, the classification accuracy of $ZeroC_2$ is 33.3%. This demonstrates that $ZeroC_2$ is able to acquire novel hierarchical concepts at *inference* time, from an *independently* trained model from a *different domain*. The ablation study shows that $L^{(em)}$, $L^{(neg)}$ and $L^{(pos)}$ all contribute to more accurate parsing. The standard technique of EBM training [12] are insufficient to achieve good parsing, leading to less accurate transfer of hierarchical concepts. We also see in the ablations that even without perfect parsing from $ZeroC_1$, the reduction of classification accuracy for $ZeroC_2$ is small, showing that it is able to classify under noisy specification of the concept graph. We discuss further on the generality (Appendix A.13), scalability (Appendix A.14) and computational complexity (Appendix A.15) in respective Appendix sections.

# 4   Related Work

Our work relates to visual compositionality, concept and relation learning, and zero-shot learning.

**Visual compositionality.** Compositionality is a key for addressing diverse tasks given finite basic knowledge. Some approaches use composition EBMs for generation [1], and a VAE-based architecture for bi-directional symbol-image generation that can also learn logical recombination of concepts [2]. These two works focus on composing *factors of variation*, *e.g.* color, position, smiling, young. In contrast, our work focuses on concepts that abstract objects, where the internal hierarchical relational structure is key. Moreover, while the above works focus on generation, our work focuses on the tasks of classification and detection. A novel Bayesian-based method for few-shot learning on the

---

[7]This is a stringent metric since, e.g. for "Eshape" that contains 4 concept nodes and 6 relation edges, an individual accuracy of 0.9 would result in $\sim 0.9^{4+6} = 0.349$ isomorphism accuracy, and an individual accuracy of 0.8 would result in $\sim 0.8^{4+6} = 0.107$ isomorphism accuracy.

Omniglot dataset shows that compositionality is pivotal for improved performance [14]. They achieve compositionality via hierarchical MCMC sampling on hand-coded priors of elementary concepts and relations. In comparison, our work is neural network-based and only requires demonstrations of elementary concepts and relations, reducing hand-coded priors. Another approach introduced a modular neural network, which uses composition of neural modules for visual question answering [15]. Their method can be seen as composing transformations on a representation. In comparison, our composition is achieved via composition of energy landscapes. While they focus on question answering on scene graphs, we focus on classification and detection of hierarchical concepts.

**Concept and relation learning.** There has been exciting progress in concept learning and relation learning. Works in concept learning generally represent concepts in latent space via prototypes [16, 17], or via latent embedding such as SCAN [2], InfoGAN [18] and Neuro-Symbolic Concept Learner [19]. [20] introduced EBMs to represent concepts with a demonstration in simple state space. [21] further introduced unsupervised learning of local and global concepts with EBMs. Regarding relation learning, non-local neural networks [22], Relation Networks [23], Neural Relational Inference [24] and C-SWM [25] use latent complete graphs as inductive biases to represent potential relations. PrediNet [3] explicitly represents propositions, relations, and objects with a transformer-based architecture, and demonstrates that it can learn relations that generalize to novel shapes of objects and column patterns. In comparison to the above works, ZeroC explicitly learns both concepts and relations, which has the unique capability to recognize and acquire hierarchical concepts at inference.

**Zero-shot learning.** Zero-shot learning methods [4, 5, 6, 7, 8] typically learn a joint embedding space between image and feature labels, and perform classification at inference time on images with novel classes, based on how the novel classes correspond to a set of features. Our generalization task also requires generalizing to new concepts without seeing the image (zero-shot), but has the important distinctions that at inference time, our concepts to be inferred lie at a *higher* hierarchy than that in training, and furthermore use the structural concept graph. In comparison, in standard zero-shot learning, concepts in training and inference lie at the *same* hierarchy level, and only generalize to new combinations of features (constituent concepts) while neglecting relation structures.

# 5    Conclusion

In this paper, we introduce ZeroC, a new framework for zero-shot concept recognition and acquisition at inference time. Our experiments show that in a challenging grid-world domain, ZeroC is able to recognize complex, hierarchical concepts composed of English characters in a grid-world in a zero-shot manner, being given a high-level, symbolic specification of their structures, and after being trained with simpler concepts. In addition, we demonstrate that an independently trained ZeroC is able to transfer hierarchical concepts across different domains at inference. Although this work is evaluated only in grid-world visual domain, we are the first to address this difficult challenge. We are also excited to see its potential application in broader domains, e.g. in AI for scientific discovery, where it may infer novel patterns and concepts from data in a zero-shot manner. We hope that this work will make a useful step in the development of composable neural systems, capable of zero-shot concept recognition and acquisition and hence suitable for more diverse tasks.

**Acknowledgments and Disclosure of Funding**

We thank Rui Yan and Blaž Škrlj for discussions and for providing feedback on our manuscript. We also gratefully acknowledge the support of DARPA under Nos. HR00112190039 (TAMI), N660011924033 (MCS); ARO under Nos. W911NF-16-1-0342 (MURI), W911NF-16-1-0171 (DURIP); NSF under Nos. OAC-1835598 (CINES), OAC-1934578 (HDR), CCF-1918940 (Expeditions), NIH under No. 3U54HG010426-04S1 (HuBMAP), Stanford Data Science Initiative, Wu Tsai Neurosciences Institute, Amazon, Docomo, GSK, Hitachi, Intel, JPMorgan Chase, Juniper Networks, KDDI, NEC, and Toshiba.

The content is solely the responsibility of the authors and does not necessarily represent the official views of the funding entities.

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
