# A Appendix

In the Appendix, we provide details that complement the main text. Specifically, in Appendix A.1 and A.2, we detail the learning algorithm of HC-EBMs and the learning objective these models optimize. In Appendix A.3, we prove a necessary conditions for ZeroC to correctly discover the underlying concept graph. In Appendix A.4, we provide the exact neural architecture used to parameterize HC-EBMs. In Appendix A.5, we present further explanation on how hierarchical concepts acquired in one domain may be transferred to different domain, as well as a note on time complexity of inference. In Appendix A.6 and A.7, we present implementation details on the CADA-VAE baseline for classifying hierarchical concepts and an analysis of its limited performance. In Appendix A.8, we similarly present implementation details on the Mask R-CNN + Relation Classification baseline for acquiring concepts. In Appendix A.9, we detail the generation process of our 2D and 3D grid-world datasets. In Appendix A.10, we explain some limitations of our current work and propose several future directions to explore. In Appendix A.11, we discuss our work's broader social impact. In Appendix A.12, we present additional experimental results on a CLEVR dataset to demonstrate the potential of our framework to generalize to more real-world settings. In Appendix A.13, we further discuss the generality of our framework to different sets of elementary concepts and relations and to other datasets. In Appendix A.14, we explain the scalability of the framework with respect to task complexity, inference time complexity, and image complexity. In Appendix A.15, we summarize ZeroC's computational complexity and our empirical observations.

## A.1 Learning algorithm

Here we give the learning algorithm for HC-EBMs, which can be elementary or hierarchical. A similar algorithm applies to R-EBMs, by simply replacing the $m$ by $m_1, m_2$. Here we omit the subscript of $X, M, C$ for clarity.

In our experiments, we perform hyperparameter search over the coefficients, to optimize the average classification accuracy and detection IoU on the validation set that has the same type of concepts/relations (not validating on the inference tasks in Section 3). For training HC-EBMs, we use $\alpha_{\text{pos-std}} = 0.1, \alpha_{\text{neg}} = 0.05$ and $\alpha_{\text{em}} = 0.1$. For training R-EBMs, we use $\alpha_{\text{pos-std}} = 1, \alpha_{\text{neg}} = 0.2$ and $\alpha_{\text{em}} = 0$. We use learning rate $10^{-4}$, number of SGLD steps $K = 60$ for training and $K = 150$ for inference. During inference, we use an ensemble size of 64 to perform MAP estimation as in Eq. (2), and use ensemble size of 256 for detection.

## A.2 Learning objective

Here we detail the learning objective for ZeroC and provide explanations, complementing Sec. 2.4. The $L_n^{\text{improved}}$ is the objective proposed in [12], neglecting the entropy regularization term. In addition, as explained in Sec. 2.4, we introduce three regularizations to inject the right inductive biases to address the tasks of zero-shot concept recognition and acquisition:

$$
\begin{cases}
L_n^{(\text{pos-std})} = \left(\text{Var}_n\big(E_\theta(x_n^+, m_n^+, c_n^+)\big)\right)^{\frac{1}{2}} \\
L_n^{(\text{em})} = \left| \frac{E_\theta(x_n^+, m_n^+, c_n^+) + E_\theta(x_n^+, m_n^-, c_n^+)}{2} - E_\theta(x_n^+, m_n^{(\text{em})}, c_n^+) \right| \\
L_n^{(\text{neg})} = -E_\theta(x_n^+, m_n^{(\text{neg})}, c_n^+)
\end{cases}
\tag{5}
$$

The superscript "+" denotes positive examples from ground-truth, "−" denotes negative examples generated from SGLD, "(em)" denotes an empty mask, and "(neg)" denotes algorithmically generated negative examples. The additional regularizations are $L_n^{(\text{pos-std})}$ for reducing energy variance for positive examples, $L_n^{(\text{em})}$ for empty-mask regularization, and $L_n^{(\text{neg})}$ which is algorithmically generated negative examples. Note that unlike standard image-only EBMs, the training needs to take multiple modalities into account: image $x$, mask $m$ and concept $c$. We perform conditional SGLD w.r.t. each modality to generate negative examples. *E.g.*, when generating negative examples for masks $m_n^-$, we provide ground-truth of image and concept label $x_n^- \leftarrow x_n^+, c_n^- \leftarrow c_n^+$. Thus, $(x_n^-, m_n^-, c_n^-) = (x_n^+, m_n^-, c_n^+)$ is a negative tuple even though $x_n^-$ and $c_n^-$ are from ground-truth. For each minibatch we randomly sample the modality the conditional SGLD is performed on (steps 4-21 in Alg. 2).

---

**Algorithm 2** Algorithm for learning HC-EBMs

---

**Require:** data dist. $p_D(x, m, c)$, HC-EBM $E_\theta$,
**Require:** step size $\lambda$, number of steps $K$, random mask generator $\mathcal{U}_a$, random embedding generator $\mathcal{U}_c$, coefficients $\alpha_{\text{pos-std}}$, $\alpha_{\text{neg}}$, and $\alpha_{\text{em}}$.
1: $\mathcal{B} \leftarrow \varnothing$
2: **while** not converged **do**
3: $\quad (x_n^+, m_n^+, c_n^+) \sim p_D$
$\quad$ *// Generate samples from Langevin dynamics:*
4: $\quad$ rand $\sim U[0, 1]$
5: $\quad$ **if** rand $\in [0, 1/4)$ **do**
6: $\quad\quad (x_n, \tilde{m}_n^0, c_n^+) \sim \mathcal{B}$ with 20% probability and $\tilde{m}_n^0 \sim \mathcal{U}_a$ otherwise
7: $\quad\quad \tilde{m}_n^K \leftarrow \mathbf{SGLD}_{\tilde{m}}(E_\theta; x_n, \tilde{m}_n^0, c_n^+)$
8: $\quad\quad (x_n^-, m_n^-, c_n^-) \leftarrow (x_n, \tilde{m}_n^K, c_n^+)$
9: $\quad$ **elseif** rand $\in [1/4, 1/2)$ **do**
10: $\quad\quad (x_n, m_n^+, \tilde{c}_n^0) \sim \mathcal{B}$ with 20% probability and $\tilde{c}_n^0 \sim \mathcal{U}_c$ otherwise
11: $\quad\quad \tilde{c}_n^K \leftarrow \mathbf{SGLD}_{\tilde{c}}(E_\theta; x_n, m_n, \tilde{c}_p^0)$
12: $\quad\quad (x_n^-, m_n^-, c_n^-) \leftarrow (x_n, m_n^+, \tilde{c}_n^K)$
13: $\quad$ **elseif** rand $\in [1/2, 3/4)$ **do**
14: $\quad\quad (x_n, \tilde{m}_n^0, \tilde{c}_n^0) \sim \mathcal{B}$ with 20% probability and $\tilde{m}_n^0 \sim \mathcal{U}_m, \tilde{c}_n^0 \sim \mathcal{U}_c$ otherwise
15: $\quad\quad \tilde{m}_n^K, \tilde{c}_n^K \leftarrow \mathbf{SGLD}_{\tilde{m}, \tilde{c}}(E_\theta; x_n, \tilde{m}_n^0, \tilde{c}_n^0)$
16: $\quad\quad (x_n^-, m_n^-, c_n^-) \leftarrow (x_n, \tilde{m}_n^K, \tilde{c}_n^K)$
17: $\quad$ **else do**
18: $\quad\quad (\tilde{x}_n, \tilde{m}_n^0, \tilde{c}_n^0) \sim \mathcal{B}$ with 20% probability and $\tilde{x}_n^0 \sim \mathcal{U}_x$ otherwise
19: $\quad\quad \tilde{x}_n^K \leftarrow \mathbf{SGLD}_{\tilde{x}}(E_\theta; \tilde{x}_n^0, m_n, c_n)$
20: $\quad\quad (\tilde{x}_n^-, m_n^-, c_n^-) \leftarrow (\tilde{x}_n^K, m_n, c_n)$
21: $\quad$ **end if**
$\quad$ *// Optimize objective for $E_\theta$ wrt $\theta$ with Eq. 4:*
22: $\quad \Delta\theta \leftarrow \nabla_\theta \frac{1}{N} \sum_n \left( L_n^{(\text{Improved})} + \alpha_{\text{pos-std}} L_n^{(\text{pos-std})} + \alpha_{\text{em}} L_n^{(\text{em})} + \alpha_{\text{neg}} L_n^{(\text{neg})} \right)$
23: $\quad$ Update $\theta$ based on $\Delta\theta$ using Adam optimizer
24: $\quad \mathcal{B} \leftarrow \mathcal{B} \cup (x_n^-, m_n^-, c_n^-)$
25: **end while**
26: **return** $E_\theta$

---

---

**Algorithm 3** Stochastic Gradient Langevin Dynamics (SGLD)

---

**Require:** energy-based model $E_\theta$ with concept (relation) embedding $c$ ($r$)
**Require:** SGLD target $\tilde{q}$, choose from $\tilde{m}, \tilde{c}, (\tilde{m}, \tilde{c})$ or $x$
**Require:** Input $x$
**Require:** step size $\lambda$, number of steps $K$
1: **for** $k = 1$ to $K$ **do**
2: $\quad \tilde{q}^k \leftarrow \tilde{q}^{k-1} - \frac{\lambda}{2} \nabla_q E_\theta(x; \tilde{q}^{k-1}) + \epsilon^{k-1}$, where $\epsilon^k \sim N(0, \sigma^2), \sigma^2 = \lambda$
3: **end for**
4: **return** $\tilde{q}^K$

---

**Reducing variance of energy for positive examples.** The use of only contrastive divergence in typical EBM training, *i.e.* pushing down energy for positive examples and pushing up energy for negative examples, is insufficient, since the composed HC-EBM needs to discover the masks of *all* its constituent models. For example, with 2 objects in the image $x^+$ with concepts $c_1^+$ and $c_2^+$, respectively, we can use $E(x^+, m_1, c_1^+) + E(x^+, m_2, c_2^+)$ according to our composition rule (Def. 2.1) to detect their respective masks. However, $L^{(\text{improved})}$ only encourages $(x^+, m_1^+, c_1^+)$ to be lower than $(x^+, m_1^-, c_1^+)$ locally, but it can still be higher than $(x^+, m_2^-, c_2^+)$ for a negative mask $m_2^-$ for concept $c_2^+$. Then the composed energy model will favor discovering $m_2^-$ instead of $m_1^+$. Thus, we add $L^{(\text{pos-std})}$ to encourage similar energy for positive examples, thus lower than the energy for negative examples.

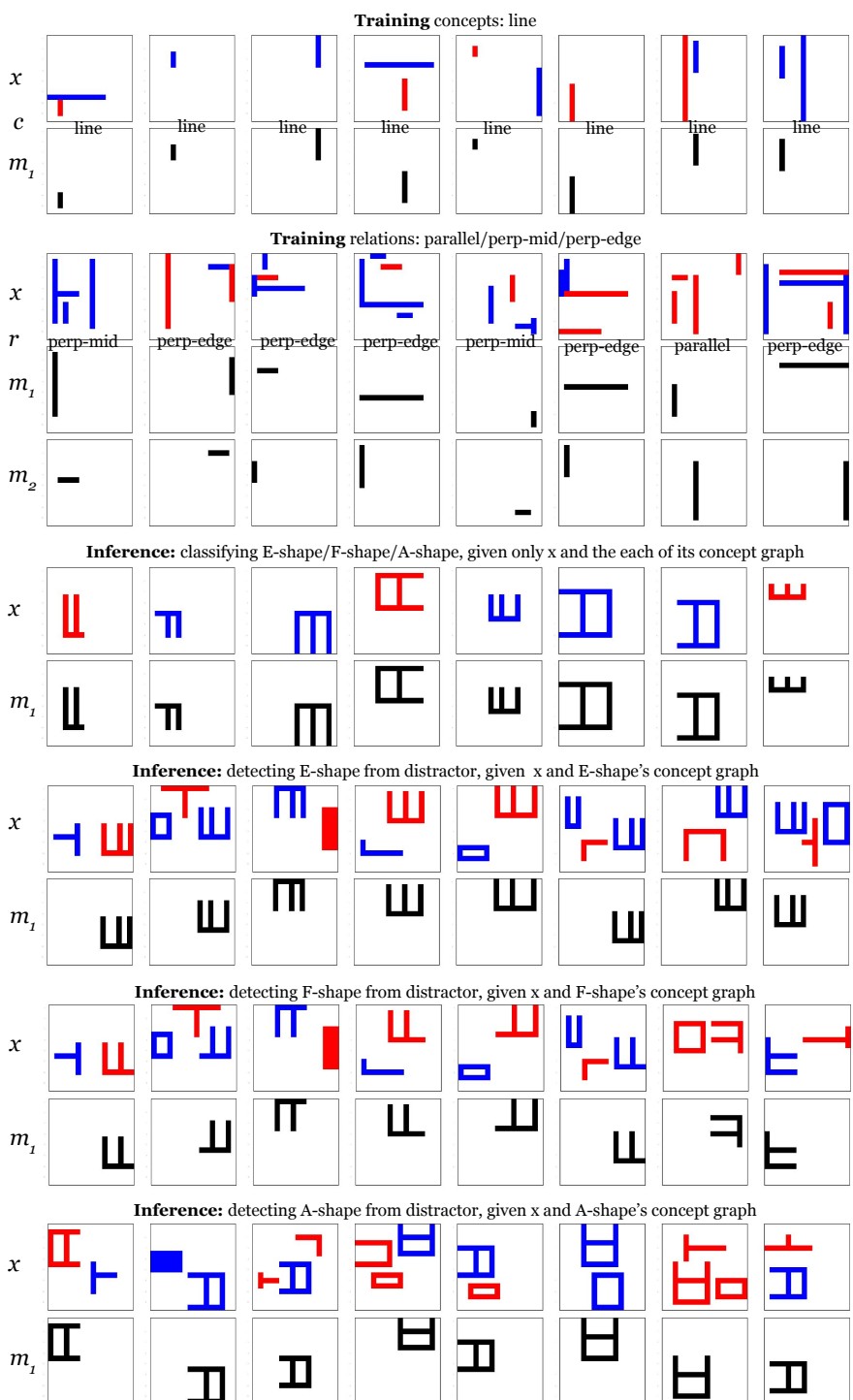

Figure 4: Samples examples from the HD-Letter dataset for training and inference of zero-shot concept classification and detection. More detail in Appendix A.9. The models are trained with concept of "line" and relations of "parallel", "perp-mid", "perp-edge". At inference, the models are tasked to perform classification and pixel-wise detection, on hierarchical concepts (w.r.t. trainnig) of "E-shape", "F-shape" and "A-shape". We see that the concepts in inference is more complex than those in training. *E.g.* for detecting "Eshape" in inference, a model will need to compose up to 4 nodes (of "line" concepts) and 6 edges of "parallel", "perp-mid", "perp-edge" relations.

**Empty-mask regularization.** To properly perform parsing (Alg. 1) when there are more energy terms than actual instances, we require that the redundant masks are empty instead of being some random negative masks. in Appendix A.3, we prove that the necessary condition for Alg. 1 to correctly discover the underlying concept graph is that the energy of the empty mask lies between the energy of positive examples and negative examples. Intuitively, given the image $x_n^+$ and concept $c_n^+$, the energy $E(x_n^+, m_n^{(em)}, c_n^+)$ with empty mask $m_n^{(em)}$ should be between the positive energy $E(x_n^+, m_n^+, c_n^+)$ and negative energy $E(x_n^+, m_n^-, c_n^+)$ for the empty mask to appear before random negative masks appear. Therefore, in $L_n^{(em)}$, we encourage $E(x_n^+, m_n^{(em)}, c_n^+)$ to be near the average of positive and negative energy. If there is redundant energy terms during parsing, the corresponding masks will favorably become all-zero instead of some random negative mask.

**Algorithmically generated negative examples.** To encourage each mask to discover a single concept instance instead of a partial instance, we randomly corrupt the tuple $(x_n^+, m_n^+, c_n^+)$ and push up the corresponding energy.

**Reason to neglect the entropy regularization.** The entropy term in [12] serves to increase the diversity of the generated examples. And the computation of entropy requires many examples. This is fine in [12] since the EBM there has the form of E(x) which only needs to generate images *unconditionally*, and the entropy can be estimated using all previous generated images x. In our work, our EBM are $E(x, m, c)$ and $E(x, m_1, m_2, c)$, and we need to generate the mask *conditionally*, e.g. generate mask m conditioned on the image $x$ and label $c$. The entropy term would need to be a conditional entropy of $m$ given $x$ and $c$, where the pool of mask $m$ should be different for each individual image $x$ and label $c$. This requires, e.g. for each $x, c$, we generate over 100 masks to estimate the entropy which is computationally expensive, while currently we only need to sample 1 mask. Moreover, typically there are limited correct masks for a concept in an image, and encouraging diversity may not help the model identify the correct mask. In fact, we have tried empirically with keeping the entropy term and it results in a much worse accuracy, likely due to the above reason.

### A.3 Proof for necessary condition for correctly parsing the graph

Here we provide the proof that the energy for an empty mask needs to lie between the positive energy and negative energy, justifying the introduction of $L_n^{(em)}$ in Eq. (4). Specifically, we prove:

**Theorem A.1.** *Let $E^+ = E_\theta(x_n^+, m_n^+, c_n^+)$ be the "positive energy" for all positive examples of $(x_n^+, m_n^+, c_n^+)$, and $E^- = E_\theta(x_n^-, m_n^-, c_n^-)$ be the "negative energy" for any negative examples[8] of $(x_n^+, m_n^+, c_n^+)$. Let $E^{(em)}$ be the energy for an $(x_n^+, m^{(0)}, c_n^+)$ where $m^{(0)} = \mathbf{0}$ is an empty mask. A necessary condition for correctly discovering the underlying concept graph with Alg. 1 is that*

$$E^+ < E^{(em)} < min(E^-, E^+ + E^{(overlap)}) \tag{6}$$

*Here $E^{(overlap)}$ is the energy added in Alg. 1 to penalize the concept EBMs to discover overlapping concepts.*

*Proof.* Suppose that in the image there are in total $m \geq 1$ objects, and there are in total $n \geq 1$ concept EBMs. We want that only $m$ EBMs have their masks enabled and all the rest $n - m$ masks are empty (if $n \geq m$), *i.e.* this configuration should have the lowest total energy. In other words suppose that instead there are $k$ masks that finds the objects (if will not overlap until $k > m$, and $n - k$ remains empty or have negative masks, then it should have a higher energy:

$$m \cdot E^+ + \lfloor n - m \rfloor \cdot E^{(em)} \leq k \cdot E^+ + \lfloor k - m \rfloor \cdot E^{(overlap)} + (n - k) \cdot E^{(em)} \tag{7}$$

$$m \cdot E^+ + \lfloor n - m \rfloor \cdot E^{(em)} < k \cdot E^+ + \lfloor k - m \rfloor \cdot E^{(overlap)} + (n - k) \cdot E^- \tag{8}$$

These two expressions should hold for any $k, m$.

---

[8]Here we make the simplifying assumption that all positive energies have the same value, and all negative energies have the same value. In fact, the $L_n^{(pos\text{-}std)}$ encourages that the positive energies to be similar, to be able to discover all relevant concepts.

Setting $k = m$ in Eq. (8), we have

$$E^{(\text{em})} < E^-$$

From Eq. (7), if $n \geq k > m$, and after re-arranging terms, we have

$$E^{(\text{em})} < E^+ + E^{(\text{overlap})}$$

From Eq. (7), if $n \geq m > k$, and after re-arranging terms, we have

$$E^+ < E^{(\text{em})}$$

Combining the above three conditions, we have

$$E^+ < E^{(\text{em})} < \min(E^-, E^+ + E^{(\text{overlap})}) \tag{9}$$

which concludes the proof.

This justifies the $L_n^{(\text{em})} = \left| \frac{E_\theta(x_n^+, m_n^+, c_n^+) + E_\theta(x_n^+, m_n^-, c_n^+)}{2} - E_\theta(x_n^+, m_n^{(\text{em})}, c_n^+) \right| = |\frac{1}{2}(E^+ + E^-) - E^{(\text{em})}|$ in Eq. (4), where it encourages that the $E^{(\text{em})}$ stays between $E^+$ and $E^-$, and penalizes the deviation of $E^{(\text{em})}$ to $\frac{1}{2}(E^+ + E^-)$.

$\square$

## A.4 Network architecture of ZeroC

For all experiments in the paper, we use the *same* architecture of concept and relation EBMs, with the only difference being the number of input channels (10 for 2D images and 3 for 3D images). We provide in Table 5 the architecture of a HC-EBM, which consists of several ResBlocks (Table 3). Adding the residual to the final output is denoted as Skip(). When downsampling is performed, the residual is the output of two fully-connected layers applied to a flattened input image; otherwise, the residual is the input image. Chunk() splits an input vector into two equal-sized halves and expands both halves along a 2d grid with the same dimensions as the input image. The C_Embed() architecture is detailed in Table 4, with c_dim denoting the dimension of the concept embedding.

Table 3: ResBlock$(x, c)$ Architecture

| Type |
| --- |
| c_embed_1, c_embed_2 ← Chunk(C_Embed($c$)) |
| Concat($x$, c_embed_1) |
| $3 \times 3$ Conv2d, 64, Spectral Norm |
| Activation() |
| Concat(out, c_embed_2) |
| $3 \times 3$ Conv2d, 64, Spectral Norm |
| Activation() |
| Skip() |

Table 4: C_Embed Architecture

| Type |
| --- |
| Dense(c_dim, $4\cdot$ c_dim) |
| LeakyRelu(0.2) |
| Dense($4\cdot$ c_dim, $4\cdot$ c_dim) |
| LeakyRelu(0.2) |
| Dense($4\cdot$ c_dim, 4) |

## A.5 Acquiring Hierarchical Concepts

We present details about the algorithm for acquiring hierarchical concepts across models and domains. The algorithm works as follows (see also Alg. 4)

We first parse the image by decomposing it $x_1$ into a graph $\mathcal{G}_1$ of concepts and relations previously learned by HC-EBM$_1$ and R-EBM$_1$. Next, we copy $\mathcal{G}_1$ to $\mathcal{G}_2$ in domain 2. Finally, we compose a new HC-EBM$_2(c)$ using $\mathcal{G}_2$, HC-EBM$_2$ and R-EBM$_2$ (Hierarchical Composition Rule, Def. 2.1). The most complex step of this algorithm is parsing, the first step, which is described in detail in the paper (see Alg. 1).

**Note on time complexity.** Note that in Alg. 1, although the relation EBM needs to perform classification for each pair of discovered objects, which scales as $n^2$ where $n$ is the number of objects, the

Table 5: HC-EBM $E_{X,M,C}(x,m,c)$ Architecture

| Type, # Channels | Activation |
|---|---|
| Concat$(x,m)$ | (-) |
| $3 \times 3$ Conv2d, 128 | LeakyRelu(0.01) |
| ResBlock (Downsample), 128 | LeakyRelu(0.01) |
| ResBlock, 128 | LeakyRelu(0.01) |
| ResBlock (Downsample), 256 | LeakyRelu(0.01) |
| ResBlock, 256 | LeakyRelu(0.01) |
| ResBlock (Downsample), 256 | LeakyRelu(0.01) |
| ResBlock, 256 | LeakyRelu(0.01) |
| Global Average Pooling | (-) |
| Dense$() \to 1$ | (-) |

---

**Algorithm 4 Acquiring Hierarchical Concepts Between Models and Domains**

---

**Require:** HC-EBM$_1$ and R-EBM$_1$ in domain 1, HC-EBM$_2$ and R-EBM$_2$ in domain 2, with prior-learned concepts or relations in their respective domain.
**Require:** Image $x_1$ in domain 1, containing unseen hierarchical concept $c$.
1: $\mathcal{G}_1 \leftarrow$ **Parse**$(x_1; \text{HC-EBM}_1, \text{R-EBM}_1)$   // *Alg. 1*
2: $\mathcal{G}_2 \xleftarrow{\text{channel}} \mathcal{G}_1$   // *send $\mathcal{G}_1$ to domain 2*
3: HC-EBM$_2(c) \leftarrow$ **Compose**$(\mathcal{G}_2; \text{HC-EBM}_2, \text{R-EBM}_2)$
   // *Using Hierarchical Composition Rule (Def. 2.1)*

---

process is actually very fast, since we can batch all the pairs into a single minibatch, and can get the classification result with a single SGLD run, which has the same runtime as doing inference with a relation EBM on a single image.

## A.6 Implementation Details for CADA-VAE

CADA-VAE [8] learns a common space latent space for image features and class embeddings, by aligning modality specific variational autoencoders. Alignment is encouraged by adding two regularization terms to the standard VAE loss. This enables discriminative latent features to be sampled for unseen classes and a softmax classifier to be trained on top of such features.

To train VAE's in encoding and decoding features in the image modality, we require a pretrained backbone specific to our domain. We obtain this pretrained model by training train a network for predicting object masks and either the concept or relation labels in a self-supervised manner [26]. Similar to the original work, we use a ResNet-12 as the backbone, which consists of 3 residual blocks of 64, 160, 320 filters, each with $3 \times 3$ convolutions. A $2 \times 2$ max pooling operation is applied after each of the first 3 blocks. Following the blocks, we have two mask prediction heads that have the same architecture. The architecture is symmetric to the ResNet-12 backbone, with 3 blocks of 320, 160, 64 filters, each with $3 \times 3$ convolutions and an upsampling layer. For classification, a global average pooling is applied after the last block. Additionally, a 4 neuron fully-connected layer is added after the final classification layer. After training the network end-to-end, we use the ResNet backbone as a pretrained feature extractor.

Class embeddings for the Hierarchical Concept corpus consist of slots for each atomic concept and relation. The number of slots per concept / relation is equal to the maximum number of times it can appear in a hierarchical concept. A single slot assignment (setting the value of a slot from 0 to 1) corresponds to an instance of the slot's concept / relation. Multiple slots are assigned if more than one instance of the matching concept / relation is found in a class. During training, where the ground truth label exists for only one concept / relation instance, we randomly sample at each minibatch to determine which of the ground truth slot to assign. In this way, the class embedding VAE should learn encodings that are invariant to permutations of assigned and non-assigned slots.

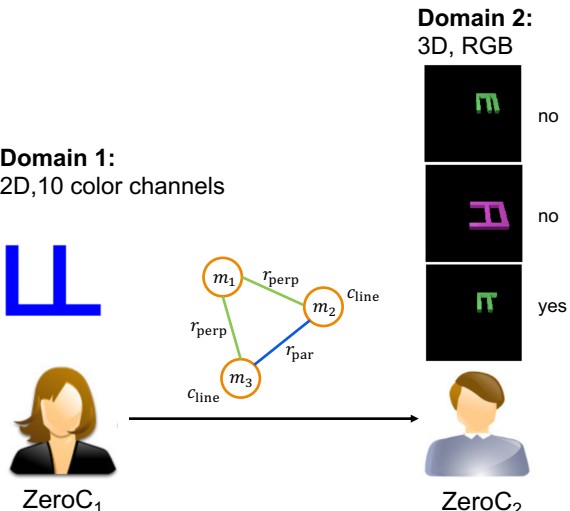

Figure 5: **Acquiring Hierarchical concepts between models and domains at inference time**. $ZeroC_1$ sees the image in domain 1. It first parses it into a structure graph using Alg. 1, then sends the graph via a communication channel to an independently trained $ZeroC_2$ in domain 2. $ZeroC_2$ can then directly classify the images in its domain.

## A.7 Investigation of CADA-VAE performance on HD-Letter dataset

In Table 1, we see that the classification accuracy of CADA-VAE is 18.0%, even lower than the "Statistics" method of 46.5%. Here we investigate the reason.

Firstly, we want to see if the image encoder in CADA-VAE has enough power to differentiate the different novel concept classes ("Eshape", "Fshape" or "Ashape") during inference. We perform t-SNE on the embeddings of test images, with each color denoting a different class (the labels are unseen to the algorithm). Fig. 6 shows the visualization. We see that the encoder is able to roughly separate the unseen images into different clusters that roughly correspond to different unseen concept classes. This shows that the image encoder of CADA-VAE has enough power and is not the reason for its low performance in HD-Letter. The reason then lies in the distribution shift of the class embedding, which is a multi-hot vector indicating the presence of individual concepts ("line") or relations ("parallel", "perp-mid", "perp-edge"). During training, the class encoder has only seen the class embeddings where only one concept or relation is present. However, during inference, a hierarchical concept (*e.g.* "Eshape") may have up to 4 concept instances (*e.g.* 4 lines) and 6 relations, thus the class embedding will have many hots activated. This constitutes a large distribution shift for the class embedding, so that CADA-VAE does not know how to interpret it. Note that this cannot be easily fixed with alternative class embedding schemes. No matter how we specify features for the classes in training and inference, the nature of our challenging HD-Letter dataset will result in a large distribution shift for the class embedding (up to 4 concept instance + 6 relation in inference vs. 1 concept/ 1 relation in training).

This shows the limitation of CADA-VAE and similar zero-shot learning methods, where they are not equipped to handle zero-shot learning of hierarchical concepts that are more complex due to the composition of learned concepts and relations. In contrast, our ZeroC naturally supports such composition into concepts with more complex structure, which enables zero-shot recognition of hierarchical concepts at inference time.

## A.8 Implementation Details for Mask R-CNN + Relation Classification

Firstly, one may note that the performance of Mask R-CNN's graph parsing accuracy (35.5%) is seemingly low. In fact, this performance is very good. Note that we use the stringent metric of isomorphism accuracy: the acc is 1 *only* if the inferred graph is isomorphic to ground-truth, and 0 otherwise. This metric presents a major challenge even for "simple" shapes. In the example of parsing the graph for "E" shape with 4 nodes (lines), 6 edges (their relations), even if each node and edge classification acc is 0.9, the isomorphism acc is $0.9^{10} = 0.349$, and 0.8 individual acc would

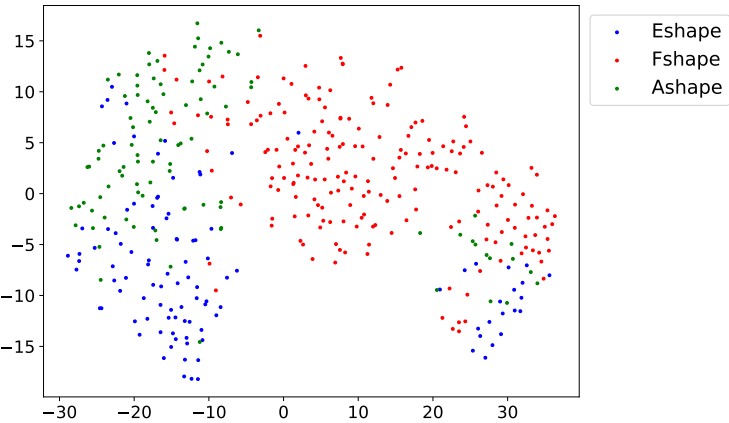

Figure 6: t-SNE visualization for the embedding of test images consisting of hierarchical concepts of "Eshape", "Fshape" or "Ashape". We see that the image encoder is able to roughly cluster the unseen concepts into different clusters.

result in $0.8^{10} = 0.107$. Thus to reach 35.5% isomorphism accuracy, a model has to have very high accuracy for classifying individual concepts and relations.

We perform intensive hyperparameter tuning for the baseline Mask R-CNN + relation classification. We first pretrain a Mask R-CNN [13] on the Hierarchical Concept corpus to minimize object segmentation and object classification losses. We then fix the Mask R-CNN weights and use the pretrained networks to output all objects in an image. Given the Mask R-CNN output for an image, the relation head is trained to minimize the relation classification loss with respect to a ground truth pair of objects and their corresponding relation.

**Mask R-CNN**: Instance segmentation is the task of precisely detecting objects through bounding-box localization and precisely segmenting each object instance while correctly predicting its corresponding class. We make use of a popular choice for instance segmentation, Mask R-CNN, which consists of a branch for predicting segmentation masks within Region of Interests (RoI) in parallel with a branch for classification and bounding box regression. The architecture for the segmentation branch consists of four convolutional upsampling blocks, each with 256 filters.

A vanilla Mask R-CNN architecture with a ResNet50 and Feature Proposal Network (FPN) [27] backbone achieves strong object localization and segmentation results on the COCO dataset [28] when trained from scratch. As COCO is much larger in scale, in terms of image resolution, number of object instances, and number of object classes, we modify the vanilla architecture to suit the Hierarchical Concept corpus's smaller image size. We use a ResNet18-FPN backbone for feature extraction, with the FPN having anchor boxes with side lengths of 4, 8, 16, 32. During training, we consider the top 2000 object proposals and reduce these to the top 1000 through NMS with a IoU threshold of 0.7. During inference, the top 20 object proposals are kept before and after NMS. We train on one GPU with a batch size of 2 for 176k iterations (equivalent to 8 epochs), with a learning rate of 0.0025.

To tune the standard Mask R-CNN architecture that works on much larger image resolutions, we decreased the size of the backbone ResNet, decreased the size of anchor boxes, and increased the scales for the RoIPooler relative to the input image. We found that decreasing the number of proposals before NMS did not improve performance, nor did decreasing the size of the mask upsampling head. We also tuned the learning rate across values in the set {0.005, 0.001, 0.0025, 0.05}, with 0.0025 yielding the best performance. Our Mask R-CNN performance for detecting lines on a subset of 1000 training images is 97.4 mIoU. We found that this performance is not significantly affected by increases in learning rate warmup length or by increases in the number of iterations before learning rate decay.

**Relation Head:** The architecture of the relation classifier consists of three residual blocks, followed by a fully-connected network with two hidden layer. Each residual block consists of 3x3 convolutions with spectral normalization, followed by downsampling. The relation head predicts the relation between all pairs of object masks outputted by the Mask R-CNN. To train the relation head, we obtain

| Concept | Description |
|---:|---|
| line | an object represents a solid line |
| rectangle | an object represents a hollow rectangle |
| rectangleSolid | an object represents a solid rectangle |
| L-shape | an object represents a shape that is "L" like |
| C-shape | an object represents a shape that is "C" like |
| A-shape | an object represents a shape that is "A" like |
| E-shape | an object represents a shape that is "E" like |
| T-shape | an object represents a shape that is "T" like |
| rand-Shape | an object that is randomly constructed |

| Relation | Description |
|---:|---|
| inside | object x is inside of object y |
| enclose | objects x is enclosed by object y |
| parallel | objects x is parallel with object y |
| perp-mid | object x is perpendicular with object y, and touch object y in its middle. |
| perp-edge | object x is perpendicular with object y, and touch object y in its edge. |
| non-overlap | objects x and y are not overlapped in both x and y axes. |

Table 6: Supported primitive concepts and relations in our Hierarchical-Concept corpus.

the predicted object masks that are closest (in terms of IoU) to the pair of ground truth masks and compute the loss on the predicted relation for the corresponding Mask R-CNN object masks. We tuned the learning rate of the relation head across values in the set {5e-5, 2.5e-5, 1e-5} and found that 2.5e-5 gave the best performance. We found that fixing the Mask R-CNN weights was essential to stable training of the relation head. The accuracy of our relation classifier is 94.5% on the training set.

## A.9 The Hierarchical Concept Corpus

In this section, we describe our Hierarchical-Concept corpus for our experiments and how it is generated. Our data generation framework is designed for generating large-scale datasets for concept and relation learning in a grid-world setting. Specifically, it samples pixel-level arbitrary objects, and places on to grid-world with predefined relations between objects. The task format is inspired by the Abstract Reasoning Corpus (ARC) proposed by [9].

Figure 4 to Figure 8 shows examples for concepts and relations used in our experiments, for the HD-Letter and HD-Concept datasets, and a dataset of 2D to 3D concept transfer that tests acquiring hierarchical concepts between domains at inference time. For both HD-Letter and HD-Concept, there are 44000 examples of concepts, split 10:1 for train and validation, and 44000 examples for relations, split 10:1 for train and validation. At inference time for hierarchical concepts, the classification task has 200 examples, and detection task has 600 examples, 200 for each of the hierarchical concepts. The 2D to 3D transfer dataset has 200 tasks. We define our primitive objects and relations in Table 6.

To generate the datasets[9], our engine consists of the following components:

**Concept (Object)**  As shown in Table 6, we define several shape primitives such as line, reactangle, L-shape, and random shape. Our data generation framework allows configurations such as color, width, height and orientation. We allow nine different colors, and four different orientations at maximum. In addition, we have composite objects as well. For instance, the "Lshape" consists of two lines with a fixed relation between them. We evaluate our models with these composite objects during inference time to evaluate their performance on recognizing composite objects based on primitive concepts.

---

[9]The code for generating the dataset can be found at project website http://snap.stanford.edu/zeroc/.

**Relation**    As shown in Table 6, we define several relation primitives such as "inside", "enclose", and "parallel". We use these relations to define the spatial relations between objects. For example, "inside" means one object is inside of the other object. A pair of objects may formulate multiple relations between them. Likewise, a pair of objects may be *unrelated* as well giving there is no primitive relation between them.

**Canvas**    Each of our examples in the corpus consists a canvas, where objects are positioned with relations between them. Our canvas is a $n \times n$ grid world, where each pixel in the grid world is a colored pixel. Our data generation framework can place objects onto the canvas with desired relations. For example, our framework can generate two objects where one of them is inside of the other. In the process, our framework samples a rectangle, and sample another object to be placed inside the rectangle. In the meantime, the framework allows multiple object pairs to be defined when placing objects. In addition, the framework allows objects to be specified with pre-defined attributes including shape, color, width, height and orientation. We also allow other configurations such as whether we allow objects to touch each other, and whether objects have unified color.

**Generation Process and Artifacts**    To generate training and evaluation sets for our experiments, we specify configurations of the canvas and repeatedly randomly generate canvas till we have enough examples. In addition, we allow distractor sampling, where we specify the number and the shape of distractors. Then, our data generation framework places distractors at random on the canvas. Our framework parses the canvas and the relations between objects after all objects are placed onto the canvas. Notice that the set of the relations between objects after the placement is a superset of the pre-defined configuration when creating the canvas. For example, if we sample two objects sharing the same color along with another random distractor. The relation between the distractor and one of the object can be free-formed. We disallow non-flatten configuration when creating canvas. For example, if there are three objects to be placed, we disallow circular relations specified between them (*i.e.*, there exist a relation between each two objects). We disallow two objects touching each other share the same color. To generate 3D images, we first generate the 2D images, and then use a standard tool of povray[10] to build 3D scenes based on the contour of the 2D image. We make sure that in the 2D to 3D transfer dataset, there is no overlap between the generated 2D images and the images used to generate 3D images by using a distinct seed.

**Design of dataset HD-Concept**    The main goal of this dataset is to test whether our method can detect different relational graph structure given the same number of concept nodes. Concretely, given 2 "rectangle" concept instances and one "E-shape" instance, there are limited ways to form a compositional concept with different relational graph structures: Concept1 is where "E-shape" is not inside any of the "rectangle". Concept2 is "E-shape" and "rectangle"$_1$ are both inside the other "rectangle"$_2$, and "E-shape" not inside "rectangle"$_1$. Concept3 is that "Eshape" is enclosed by one "rectangle" which is also enclosed by another "rectangle". All three compositional concepts have the same number of constituent concepts but different relation graphs.

Example images datasets are shown in Fig. 4, Fig. 7 and Fig. 8.

## A.10   Limitations of current work

In this work, we have demonstrated the zero-shot concept recognition and acquisition capability of ZeroC, with experiments in a grid-world domain. We focus on grid-world since it provides a systematic and challenging testbed to evaluate the above two capabilities of models, similar to many other pioneering works that evaluate their models in grid-world that captures the essence of the problem (*e.g.* RL in early days [29], PrediNet [3], program synthesis evaluated with Karel dataset [30], BabyAI [31], Machine Learning Theory of Mind [32], etc.) Nevertheless, the fact that our main experiment is in grid-world is a limitation of current work. In the main Sec 3.2, we train ZeroC$_2$ with 3D images, and the fact that ZeroC$_2$ is able to zero-shot classify and detect in its domain demonstrate that ZeroC is able to handle more complex 3D images. Moreover, in Appendix A.12, we perform additional zero-shot classification with a variant of CLEVR, which shows that our ZeroC is able to handle more realistic 3D images, and out-perform strong baseline of CADA-VAE.

---

[10]https://github.com/POV-Ray/povray/tree/latest-stable

Another limitation of current work is that we have only considered one hierarchy of composition, without considering more hierarchies. Although more hierarchies is in principle possible with the ZeroC method, it is out of scope of this work, since in this work we focus on demonstrating *whether* zero-shot recognition and acquisition of concepts are possible with our model. It is an exciting future work to build on current work, to explore zero-shot recognition with multiple levels of hierarchies.

Our ZeroC architecture naturally supports continued expansion or compression of the EBM pool, as newly learned compositional concept EBMs can be dynamically added to the pool. Independently trained EBMs on new concepts/relations can also be added to the pool and composed together with existing EBMs. This is an exciting future direction, but is out-of-scope of the present paper, since here we focus on proposing the framework and demonstrate the zero-shot recognition and acquisition capability of ZeroC.

### A.11 Broader social impact

Here we discuss the broader social impact of our work, including its potential positive and negative aspects. On the positive side, the capability of ZeroC enables will improve the generalization capability of deep learning models, allowing them to acquire concepts and address more diverse tasks at inference time. This provides a possible method to address the long-standing problem that deep learning models has limited generalization capability and mainly learn via examples. Our ZeroC also improves interpretability of models, since we can know exactly the structure of the concepts the model learns, allowing us to know *how* the model makes such decision.

We see no obvious negative social impact of this work. In its current state, ZeroC's capability is still very limited, not nearly addressing any of the tasks as good as human level. It will only improve the interpretability and versatility of the models, which can be used to better address challenging tasks in society.

### A.12 Additional experiments

Here we perform additional experiments with a CLEVR dataset to explore whether ZeroC can generalize to more realistic images. The models are trained with basic concepts of "Red", "Cube", "Large" and relations of "SameColor", "SameShape", "SameSize" (See Fig. 9), where the models are provided with tuples of ($image$, $mask(s)$, $label$) for concept or relation. During inference, given only symbolic specification of three hierarchical concepts, the model needs to zero-shot classify whether an image contains an instance of such concept. For example, HConcept1 is defined as three objects where one object is red, the second object has the same color as the first object and the third object has the same shape as the second object. The other two hierarchical concepts have a similar structure of specification. We used 100K images for training and 200 for inference (evaluation). The table below shows the results of our model and comparison with CADA-VAE and the statistics baseline. We see that ZeroC achieves significantly higher performance on the task than the baselines, and able to zero-shot classify more realistic images.

| Model | Classification acc (%) |
|---|---|
| Statistics | 33.4 |
| CADA-VAE | 45.3 |
| ZeroC (ours) | **56.0** |

### A.13 Generality of ZeroC approach

The ZeroC framework is quite general. The generality lies in the following two folds:

**Generality of learning the elementary concepts and relations** The EBMs in the ZeroC framework can learn general primitives as long as labeled data is provided to demonstrate the concept or the relation, even if that concept or relation is a range that contains some intrinsic variation. For example, to learn the "acute angle" relation between two lines (with varying angles), ZeroC only needs a dataset that contains many (image, mask1, mask2, "accute-angle") tuples where the mask1 and mask2 identify the two lines in the image that form an acute angle, with different examples

containing different angles. In other words, as long as the dataset contains enough data that identifies a concept/relation in a certain range, the ZeroC can learn such primitives. This is also shown in the HD-Concept dataset in Section 3.1, where "inside", "outside", "non-overlap" relation primitives are learned, and each relation has intrinsic variation. For example, the two masks for "inside" relation can have different positions, relative positions, and sizes.

**Generality to different datasets and scenarios**   Our architecture is general to learn diverse concepts and relations. For all our experiments, we use the *same* network architecture (Appendix A.4), for the (1) dataset HD-Letter, (2) dataset HD-Concept (that contains more complex concepts and relations e.g. "rectangles", "Eshape", "inside", "enclose"), (3) Section 3.2 "Acquiring Novel Hierarchical Concepts Across Domains", and (4) CLEVR experiment in Appendix A.12. This shows that the algorithm is very general, not tuned toward specific concepts or scenarios. The architectures only differ in the number of input channels (since the 2D images have 10 channels and 3D images have 3 RGB channels). For future work, we can also experiment with using a single *model* to learn concepts or relations across datasets, which is out-of-scope of current work.

## A.14   Scalability of ZeroC

**Scalability to task complexity**   We have demonstrated that even for larger images like the 3D image (32x32x3) in Section 3.2 and CLEVR in Appendix A.12 (64x64x3), our approach achieves reasonable accuracy, significantly outperforming baselines. This shows the scalability of our methods to larger images and realistic use cases. In addition, downsampling the images to lower resolution can be performed to reduce the complexity of the inference and learning.

**Scalability in terms of time complexity**   In terms of time complexity, the SGLD inference algorithm (Alg. 3) uses a fixed number of iteration steps $K$. We find that $K = 60$ is enough for reasonable detection accuracy for larger images. For larger images, a single step of SGLD make take slightly longer to run due to the larger image size. For parsing hierarchical concept from image (Alg. 1), the detection of all concept instances can be obtained for a single SGLD run, and for the classification of relations, we can concatenate all pairs of concept masks into a single minibatch and feed into the relation-EBM, which only requires one relation-EBM forward run, which is instant. Thus we see that the time complexity is fairly constant for increasing number of objects in the images and larger image size, measured in terms of a single forward or SGLD step.

**Potential application to real world images**   In order to scale to recognizing real-world objects, the ZeroC architecture and pipeline will be able to do that in principle. The main bottleneck is presented by labeled data, as we need to have detailed labels for many elementary concepts and relations that constitute real world objects. For example, the CUB-200-2011 dataset [33] provides annotations for elementary concepts for birds, e.g. beak, belly, tail, etc. This dataset lacks relation annotations, so is unsuitable for our pipeline. A suitable dataset for our pipeline can be like an augmented dataset to the above CUB-200-2011 dataset that also has relation annotations like "connect-to", "up", "down", "extend", etc. With such annotations, we can learn both concept EBMs and relation EBMs and compose together to recognize compositional concepts like different species of birds.

## A.15   Computational complexity of ZeroC

The computational complexity of the inference is detailed in Appendix A.14. In summary, the run time remains fairly constant increasing number of objects in the images and larger image size, measured in terms of a single forward or SGLD step. Since learning take the inference as an inner loop, it also remains fairly constant for the same model structure. For increasing EBM model size, the increased number of parameters will definitely require more time to train, as is typical for deep learning models. Empirically, we observe that for the same model architecture, it takes similar number of epochs to learn reasonably for different datasets.

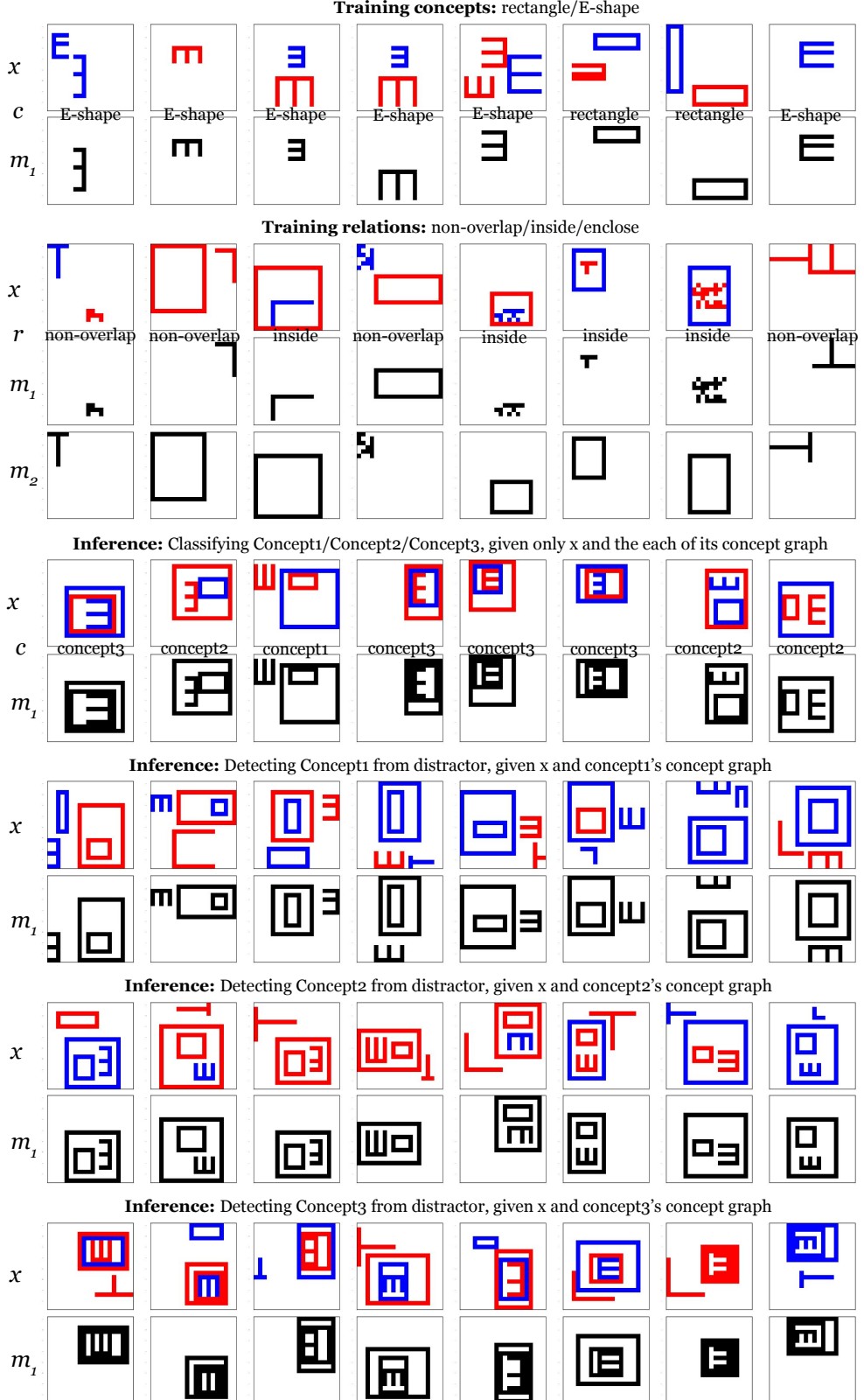

Figure 7: Samples examples from the HD-Concept dataset for training and inference. The models are trained with concept of "E-shape" and "rectangle", and relations of "non-overlap", "inside", "enclose" (inverse in "inside", if exchanging two masks). At inference, the models are tasked to perform classification and pixel-wise detection, on hierarchical concepts (w.r.t. training) of "Concept1", "Concept2" and "Concept3" (see the bottom 3 panel). In this dataset, during training relation, the pair of objects does not appear in inference, testing if the relation can generalize to new objects.

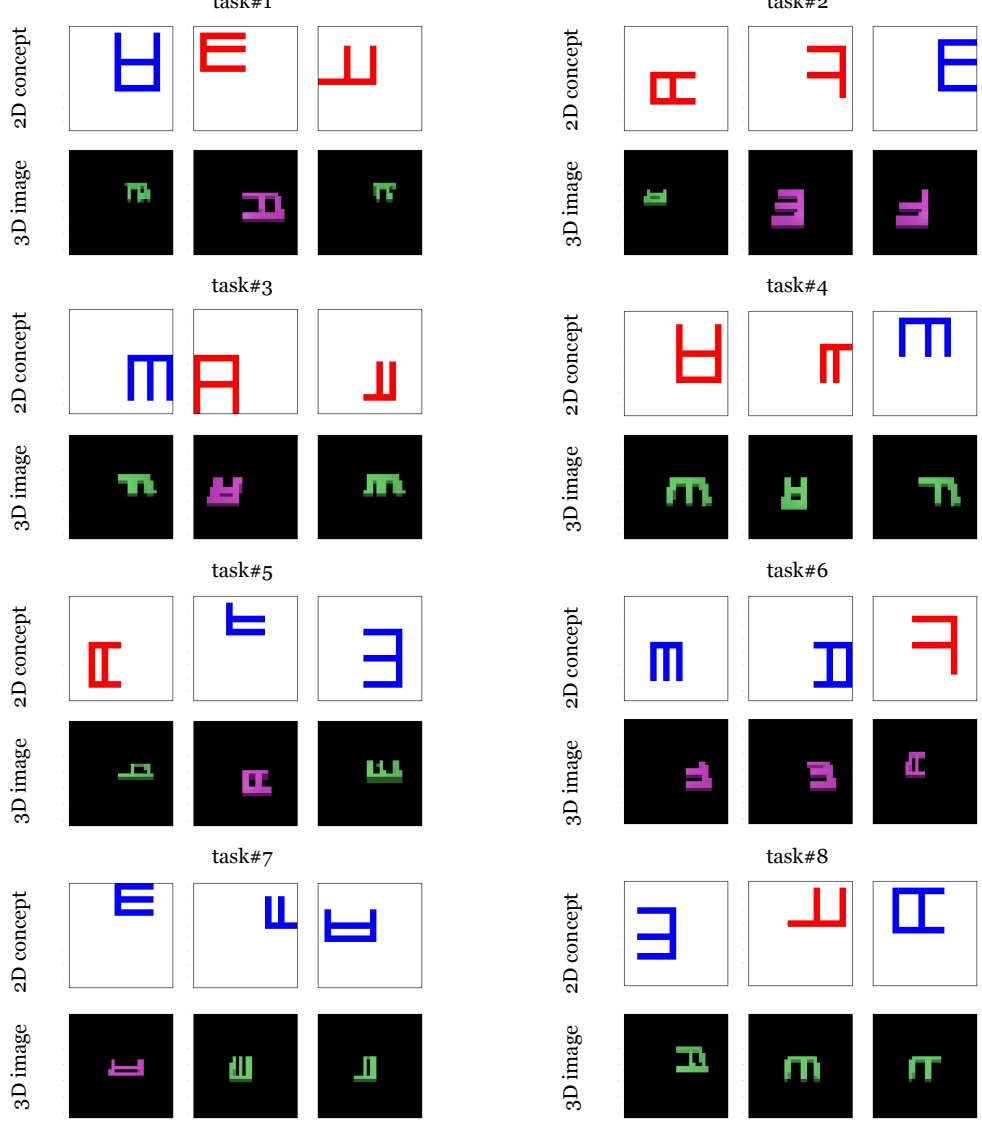

Figure 8: Examples from our Hierarchical-Concept corpus for experiments of acquiring hierarchical concepts between domains. In each panel, the upper three images demonstrate the concepts in domain 1, and after communicating high-level knowledge, independently trained models in domain 2 need to perform classification and detection on its own domain.

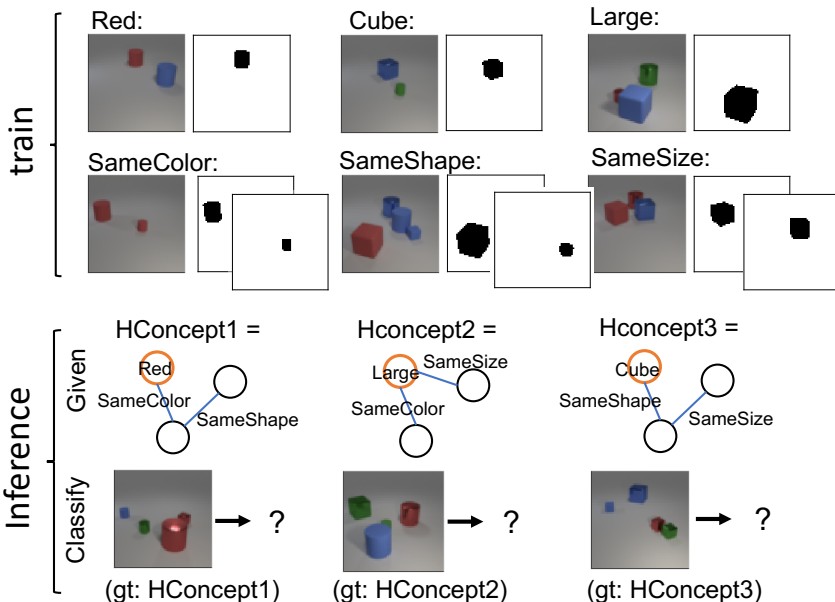

Figure 9: Zero-shot classification task based on CLEVR. Here during training, the models are given only tuples of $(image, mask(s), label)$ for elementary concepts (Red, Cube, Large) or elementary relations (SameColor, SameShape, SameSize). In inference, the models are asked to perform zero-shot classification of a hierarchical concept. For example, in lower left figure, the models are asked to classify whether an image contains HConcept1 (True if the image contains three objects where one object is red, the second object has the same color as the first object and the third object has the same shape as the second object, and False otherwise.) HConcept2 and HConcept3 have similar form of interpretation. This is a challenging task since in inference, the concepts to classify are more complicated than that in training.