# OpenReview forum: "ZeroC: A Neuro-Symbolic Model for Zero-shot Concept Recognition and Acquisition at Inference Time"
_NeurIPS.cc/2022/Conference — NeurIPS 2022 Accept_

### Official Review · Reviewer_P8en · 2022-07-08

**Rating:** 6
**Confidence:** 3
**Soundness:** 3 good
**Presentation:** 3 good
**Contribution:** 3 good

**Summary:**

This paper addresses the problem of zero-shot concept recognition. By representing concepts with a graph, and by exploiting an energy-based model, the proposed approach enables detection and classification of novel entities at inference time. The approach can be considered "neuro-symbolic" as it combines symbolic graph-representation with (neural) energy-based models with a one-to-one translation of one into the other.


**Questions:**

* There is no description of the computational cost of the proposed approach. For example, Equation 2, although it is computed via MAP estimation, and although it needs only to find the concept with the highest value in the numerator, appears to be costly. Could the authors comment on this point?


**Limitations:**

The experimental evaluation is conducted on a single artificial dataset. While this is comprehensible for a novel task, it is not clear to what extent the proposed approach is tailored on the domain of images (e.g., see also Algorithm 1) or whether it could be easily genaralized to other domains.

Also, a discussion of the computational cost of the proposed approach is missing.

Typos:
- Line 29, "which requires" -> "which require"
- Line 71, "that can given" -> "that given"
- Line 126, "to determine if" -> "to determine whether"
- Line 134, "a English" -> "an English"

**Strengths And Weaknesses:**

+ The addressed problem is challenging and important.
+ The proposed solution is sound and elegant.
+ The experimental evaluation, though on a single dataset, is well conducted.

- Generalization to multiple datasets and scenarios is not straightforward.
- No discussion on computational complexity.

---

> ### Author Response · Authors · 2022-08-02
> **Official Response**
>
> We thank the reviewer for the positive and constructive review, and glad that the reviewer recognizes the significance, soundness of the paper and well-conducted experiments. In the following, we address the limitations raise by the reviewer:
>
> > Generalization to multiple datasets and scenarios is not straightforward.
>
> > The experimental evaluation is conducted on a single artificial dataset. While this is comprehensible for a novel task, it is not clear to what extent the proposed approach is tailored on the domain of images (e.g., see also Algorithm 1) or whether it could be easily genaralized to other domains.
>
> Our architecture is general to learn diverse concepts and relations. For all our experiments, we use the <em>same<em> network architecture (Appendix A.4), for the (1) dataset HD-Letter, (2) dataset HD-Concept (that contains more complex concepts and relations e.g. “rectangles”, “Eshape”, “inside”, “enclose”), (3) Section 3.2 “Acquiring Novel Hierarchical Concepts Across Domains”, and (4) CLEVR experiment in Appendix A.12. This shows that the algorithm is very general, not tuned toward specific concepts or scenarios. The architectures only differ in the number of input channels (since the 2D images have 10 channels and 3D images have 3 RGB channels). For future work, we can also experiment with using a single <em>model<em> to learn concepts or relations across datasets, which is out-of-scope of current work. We have added the above discussion to Appendix A.13.
>
> > Also, a discussion of the computational cost of the proposed approach is missing
>
> Thanks for the suggestion! We have added Appendix A.15 in the revised version which discussed the computational cost of the proposed approach.
>
>
> > Typos
>
> Thanks for spotting the typos! We have updated them in the revised version of the draft.
>
> Above all, we hope we have resolved your concerns, which makes the paper stronger.

---

### Official Review · Reviewer_AN9q · 2022-07-14

**Rating:** 5
**Confidence:** 4
**Soundness:** 2 fair
**Presentation:** 3 good
**Contribution:** 2 fair

**Summary:**

An EBM hierarchically describing concepts and relationships is used for concept discovery in a crowded scene with irrelevant concepts. This graph can be transferred across different domains.

**Questions:**

1. Even though it is claimed that the algorithm can learn new concepts, isn't the structure of the energy model used in this paper already tuned to the discovery of spatial relationships between horizontal and vertical lines? Or if I run the same algorithm on data with arbitrary concepts (for instance, triangle, circle, object A inside objet B, object is a closed contour, etc.) will it be able to discover those as well?

2. How does inference scale if I increase the resolution to 256 x 256? Would it work? For the size and structure of your dataset, isn't the partition function of your energy computable (albeit expensive)?

3. Same question about learning

4. Isn't the particular way in which you encode spatial relationships (relative) mean that this model cannot distinguish the concept of W and M, or 6 and 9, since they are both rotations of each other, and therefore satisfy the same relative relationships?

5. This model resembles heavily the ones in [1] and [2], which also describe letters as graphs of lateral relationships that entangle nodes containing edges. What are the main differences? Can this model be used to solve CAPTCHAs? Experiments showing this would definitely be much more convincing as to its capabilities.

6. I wasn't able to understand precisely which information is conveyed from ZeroC1 to ZeroC2. Could you clarify this section in the paper?

7. When using the loss from [12], you mention that you neglect the entropy term. What's the problem with keeping it? Would the results from [12] improve had they neglected it?

Minor: Figure 4 in the appendix, training relation, first column, doesn't look like a perp-edge.


[1] A generative vision model that trains with high data efficiency and breaks text-based CAPTCHAs. Dileep George et al. Science 2017

[2] Generative Shape Models: Joint Text Recognition and Segmentation with Very Little Training Data. Xinghua Lou. NIPS 2016.

**Limitations:**

Described above

**Strengths And Weaknesses:**

Strengths:

Technically correct

- Addressing the relevant problem of hierarchical concept representation
- Clarifying examples
- Introduces a new dataset to test these ideas

Weaknesses:

- The size of the problem makes it too simple to brute force
- Unclear whether inference would scale
- Unclear how to learn arbitrary concepts from data
- Unclear how learning would scale

---

> ### Author Response · Authors · 2022-08-02
> **Official Response (continued 2)**
>
> > 4. Isn't the particular way in which you encode spatial relationships (relative) mean that this model cannot distinguish the concept of W and M, or 6 and 9, since they are both rotations of each other, and therefore satisfy the same relative relationships?
>
> Whether we can distinguish two compositional concepts that are rotations of each other depends on the primitive concepts/relations the ZeroC learns. If both primitive concepts and primitive relations are rotationally invariant (as in the dataset of the present paper), then the composed EBM is also invariant to the composed hierarchical concept (which can be easily proved in the Def. 2.1 “Hierarchical Composition Rule”). If, instead, we want to distinguish two compositional concepts that are rotations of each other, we can design the primitive concepts and/or relations such that they are not rotationally invariant. For example, we can define two “perpendicular” relations: “perp1” denotes perpendicular relation with the right angle on the upper left, and “perp2” denotes perpendicular relation with the right angle on the bottom right, in this way “6” and “9” can be distinguished.
>
> > 5. This model resembles heavily the ones in [1] and [2], which also describe letters as graphs of lateral relationships that entangle nodes containing edges. What are the main differences? Can this model be used to solve CAPTCHAs? Experiments showing this would definitely be much more convincing as to its capabilities.
>
> Compared to reference [1][2], our work differs in (1) goal: we focus on zero-shot recognition to compositional concepts, and zero-shot concept acquisition, while [1][2] focuses on recognizing CAPTCHAs in complex scenarios. (2) architecture, we use energy-based model as base model and compose them to recognize novel hierarchical concepts, while [1] uses a Recursive Cortical Network (RCN), and [2] first needs to construct a Generative Shape Model for the fonts, then   parse factor graph by solving an optimization problem. Our ZeroC requires much less engineering effort to adapt to the specific dataset, and can learn more general concepts and relations as explained in the answer to question 1. (3) Learning: we use contrastive divergence for learning the EBMs, while RCN in [1] is learned in a bottom-up way, and [2] uses a maximum-margin structured output learning paradigm.
>
> This model in principle is able to solve CAPTCHAs. It will be an exciting future work.
>
> > 6. I wasn't able to understand precisely which information is conveyed from ZeroC1 to ZeroC2. Could you clarify this section in the paper?
>
> The information conveyed from ZeroC1 to ZeroC2 is the graphical structure of a hierarchical concept. For example, in Figure 3, ZeroC1 learns the graphical structure of an E shape in terms of the initial concepts and relations. The graph structure is then conveyed to ZeroC2, which enables it to classify and detect E shapes in the 3D domain.
>
> > 7. When using the loss from [12], you mention that you neglect the entropy term. What's the problem with keeping it? Would the results from [12] improve had they neglected it?
>
> The entropy term [12] serves to increase the diversity of the generated examples. And the computation of entropy requires many examples. This is fine in [12] since the EBM there has the form of E(x) which only needs to generate images <em>unconditionally<em>, and the entropy can be estimated using all previous generated images x. In our work, our EBM are E(x,m,c) and E(x,m1,m2,c), and we need to generate the mask <em>conditionally<em>, e.g. generate mask m conditioned on the image x and label c. The entropy term would need to be a conditional entropy of m given x and c, where the pool of mask m should be different for each individual image x and label c. This requires, e.g. for each x, c, we generate over 100 masks to estimate the entropy which is computationally expensive, while currently we only need to sample 1 mask. Moreover, typically there are limited correct masks for a concept in an image, and encouraging diversity may not help the model identify the correct mask. In fact, we have tried empirically with keeping the entropy term and it results in a much worse accuracy, likely due to the above reason. We have added this discussion to Appendix A.2 of the paper.
>
> > Minor: Figure 4 in the appendix, training relation, first column, doesn't look like a perp-edge.
>
> Thanks for spotting it. It was a typo when we made the figure. We have corrected it in the revised version.

---

> > ### Author Response · Authors · 2022-08-08
> > **Could you review our response and update your rating?**
> >
> > Thank you for your review! In our response, we have addressed your concerns about scalability (of learning and inference), generality and answered your questions. In the revised version of the paper, we have also added Appendix A.13 for generality, A.14 for scalability, and augment A.9 explain about the entropy term.
> >
> > We would be grateful if you could reply to our rebuttal if you have any more questions. If our answers were satisfactory for your concerns, we would be extremely grateful if you consider updating your rating. Thank you!

---

> ### Author Response · Authors · 2022-08-02
> **Official Response (1)**
>
> We thank the reviewer for the review. Below we address the reviewer’s concerns and questions.
>
> **Under “Weakness”**:
> > The size of the problem makes it too simple to brute force
>
> We respectfully disagree. Firstly, in our work, we address the <em>relative<em> difficulty between training and test dataset, where we need to classify and detect hierarchical concepts in the test dataset that are more difficult than in training. Secondly, the detection/classification task itself is not simple, it has to solve a NP-hard subgraph-isomorphism problem. For example, as stated in the paper, to detect Eshape (a hierarchical concept with 4 nodes (lines), 6 edges (relations)) in an image with distractors of a rectangle and a “T” (this image has an underlying graph of 10 nodes (lines), 45 possible edges), we will have $C_{10}^4 \times 4!=5040$ possible mask assignments. Additionally, a model may not perfectly detect the masks for low-level concepts. Lastly, many of the detection tasks are not trivial, for example detecting “line” concept when the line is connected or crossed with other lines, detecting more complex relations of “inside”, “encompass” in HD-Concept dataset in Section 3.1, detecting “perpendicular” concept in 3D scene images in Section 3.2, and detecting more complex concept and relations in the CLEVR experiment in Appendix A.12.
>
> > Unclear whether inference would scale
> > Unclear how learning would scale
>
> We have demonstrated that even for larger images like the 3D image (32x32x3) in Section 3.2 and CLEVR in Appendix A.12 (64x64x3), our approach achieves reasonable accuracy, significantly outperforming baselines. This shows the scalability of our methods to larger images, and as reviewer maWL puts it, “suggest possible applicability to realistic use cases: 2D-to-3D domain adaptation, and generalization to real-world images”. In terms of time complexity, the SGLD inference algorithm (Algorithm 3) uses a fixed number of iteration steps K. We find that K=60 is enough for reasonable detection accuracy for larger images. For parsing hierarchical concept from image (Algorithm 1), the detection of all concept instances can be obtained for a single SGLD run, and for the classification of relations, we can concatenate all pairs of concept masks into a single minibatch and feed into the relation-EBM, which only requires one relation-EBM forward run, which is instant. Thus the inference can scale to more complex images and also images with more complex relations. For more, please see the response to the second question in the following. We have added the above discussion to the Appendix A.14 of the revised paper.
>
> > Unclear how to learn arbitrary concepts from data (in “Limitation”)
>
> > 1. Even though it is claimed that the algorithm can learn new concepts, isn't the structure of the energy model used in this paper already tuned to the discovery of spatial relationships between horizontal and vertical lines? Or if I run the same algorithm on data with arbitrary concepts (for instance, triangle, circle, object A inside objet B, object is a closed contour, etc.) will it be able to discover those as well? (in “questions”)
>
> Our model can work with any initial set of concepts and relations. For all our experiments, we use the <em>same<em> network architecture (Appendix A.4), for the (1) dataset HD-Letter, (2) dataset HD-Concept (that contains more complex concepts and relations e.g. “rectangles”, “Eshape”, “inside”, “enclose”), (3) Section 3.2 “Acquiring Novel Hierarchical Concepts Across Domains”, and (4) CLEVR experiment in Appendix A.12. This shows that the algorithm is very general, not tuned toward specific concepts. It is definitely able to detect concepts/relations of triangle, circle, object A inside object B, object is a closed contour as the reviewer suggests should such training data is provided.
>
> > 2. How does inference scale if I increase the resolution to 256 x 256? Would it work? For the size and structure of your dataset, isn't the partition function of your energy computable (albeit expensive)?
> > 3. Same question about learning
>
> We haven’t tried with resolution of 256x256, but we have demonstrated that our inference and learning can achieve reasonable performance for larger images like the 3D image (32x32x3) in Section 3.2 and CLEVR in Appendix A.12 (64x64x3). Scaling to larger images will be left for future work. In addition, downsampling the images to lower resolution can be performed to reduce the complexity of the inference and learning.
>
> For the size and structure of the dataset, the partition function of the energy is not computable, since it needs to sum over all configurations of masks. For an 16x16 image, the total number of configuration of masks is $2^{16\times16}=10^{77}$ (similar to the number of atoms in the Universe), and for an 32x32 image, the total number of configuration of masks is $2^{32\times32}=10^{308}$. They are clearly not computable.

---

### Official Review · Reviewer_Yv6P · 2022-07-18

**Rating:** 5
**Confidence:** 3
**Soundness:** 2 fair
**Presentation:** 2 fair
**Contribution:** 2 fair

**Summary:**

This work proposes a framework for zero-shot concept learning, i.e., learning and recognizing new concepts at inference time. The proposed frameworks extracts concepts and relations and composes them to learn new concepts hierarchically. The proposed method is evaluated on some synthetic datasets.


**Questions:**

How would the basic relations and concepts be defined or discovered for real world images?

**Limitations:**

It seems non-trivial to apply the current approach successfully to real world images, which is the main limitation and the authors might want to have more discussion about it and future works to address remaining challenges.

**Strengths And Weaknesses:**

Strength

(1) Zero-shot concept learning at inference time is an interesting problem and requires more attention.

(2) This work proposed a neat framework for learning concepts in a compositional way using energy based models and hierarchical composition.

Weakness

(1) There should be more discussion about how the proposed approach can scale to real world images that can include many edges and shapes with different relations. It seems this might become prohibitively costly.

(2) The evaluation would be more convincing if it can also include results on some few-shot learning datasets such omniglot or, even better, some real world images.

---

> ### Author Response · Authors · 2022-08-02
> **Official Response**
>
> We thank the reviewer for the review. Below we address the reviewer’s concerns and questions.
>
> Under “Weakness”:
> > (1) There should be more discussion about how the proposed approach can scale to real world images that can include many edges and shapes with different relations. It seems this might become prohibitively costly.
>
> In fact, in the original submission, we have conducted 2 experiments that test our approach on more realistic images. One is the CLEVR dataset with 3D scene images (see Appendix A.12). We show that our method significantly outperforms the CADA-VAE and the statistics baseline. The other is the Section 3.2 of 2D-to-3D domain adaptation, where the second domain contains 3D scene images. As Reviewer maWL puts it, the above experiments “suggest possible applicability to realistic use cases: 2D-to-3D domain adaptation, and generalization to real-world images.” We would like to emphasize here that the main aim of the paper is the introduction of a novel framework that, for the first time, demonstrates zero-shot hierarchical concept recognition, and acquisition of such concepts, even across domains. We use a grid-world dataset for systematic evaluation as no current dataset provides suitable configurations, and also have 2 experiments with more realistic 3D images showing that our framework can handle more difficult scenarios. Scaling to real world images is an exciting next step, which will be our future work.
>
>
> > (2) The evaluation would be more convincing if it can also include results on some few-shot learning datasets such omniglot or, even better, some real world images
>
> Other existing datasets lack annotations for primitives concepts and/or relations that are necessary to discover complex concepts at inference time. For example, Omniglot lacks relation annotations, and thus unsuitable for our tasks. Note that our setting is a zero-shot learning setting, different from typical few-shot learning setting. In the response to the previous question, we have explained that our submission already contains two experiments that, as Reviewer maWL puts it, “suggest possible applicability to realistic use cases”. Scaling to real world images will be an exciting future work.
>
> > How would the basic relations and concepts be defined or discovered for real world images?
>
> For real world images, we will still use the same pipeline, and in the data we need to provide elementary concept and relation annotations for real world images. For example, the CUB-200-2011 dataset [1] provides annotations for elementary concepts for birds, e.g. beak, belly, tail, etc. This dataset lacks relation annotations, so is unsuitable for our pipeline. A suitable dataset for our pipeline would be like an augmented dataset to the above CUB-200-2011 dataset that also has relation annotations like “connect-to”, “up”, “down”, “extend”, etc. With such annotations, we can learn both concept EBMs and relation EBMs and compose together to recognize compositional concepts like different species of birds. We have added the above discussion to the Appendix A.14 “Scalability of ZeroC” in the revised paper.
>
> [1] Catherine Wah, Steve Branson, Peter Welinder, Pietro Perona, and Serge Belongie. The Caltech- UCSD Birds-200-2011 dataset. 2011

---

> > ### Author Response · Authors · 2022-08-08
> > **Could you review our response and update your rating?**
> >
> > Thanks very much for you review! In our response, we have attempted to addressed your concerns about scalability, few-shot learning datasets and question about learning real-world concepts. We have also added Appendix A.14 in the revised version to discuss about scalability, which our 2D to 3D domain adaptation experiment and 3D CLEVR experiments show.
> >
> > We would be grateful if you could reply to our rebuttal if you have any more questions. If our answers were satisfactory for your concerns, we would be grateful if you consider updating your rating. Thank you!

---

### Official Review · Reviewer_maWL · 2022-07-19

**Rating:** 6
**Confidence:** 4
**Soundness:** 4 excellent
**Presentation:** 3 good
**Contribution:** 3 good

**Summary:**

The main contribution of this paper is a system (ZeroC) for pattern detection and classification that, once trained on elementary concepts, is able to zero-shot adapt to complex concepts as long as the complex concept can be described as a composition of elementary concepts & relations via a graph. Individual energy-based models (EBMs) can be trained to match each elementary concept or relation, each assigning low energy to patterns that fit its ‘signature’ (i.e. template). To recognize a complex pattern, individual EBMs can be composed according to the concept-composition graph, which allows for zero-shot recognition for arbitrary combinations of learned concepts and relations.

To evaluate zero-shot concept recognition under the scenario that the proposed method is designed for, this paper also introduces two datasets since no existing dataset are sufficiently relevant. One dataset is easier, containing letters and simple relations, while the other is harder, containing more complex patterns and relations.

Experiments show that ZeroC outperforms existing zero-shot concept recognition approaches on the proposed datasets, in both in-domain and cross-domain (2D images $\rightarrow$ 3D images) settings. Additional experiments on CLEVR give clues about potentially broader usage.

**Questions:**

Here are a handful of questions pertaining to experimentation details that could potentially make the paper more insightful.
- Line126: “*We want to determine if the concept c appears in a given image x. We need to marginalize over the mask m.*” How sensitive is your approach to the **tightness** of a mask? For example, does the concept c have to be fully occupying the entire masked space to be recognized?
- Line 149: “*the mask M is the maximum of all the masks*”. I suppose each constituent is being recognized independently, right? Then is it possible that the predicted masks don’t form a contiguous region, which would indicate a detection error? How would you plan to address this case?
- Line 230: “*The hierarchical concepts to be classified and detected are three characters which we term Concept1, Concept2, Concept3*”. These three concepts look like random patterns. How did you choose them in the first place?
- Does your approach exhibit any flexibility in recognizing concepts that have been slightly stretched or squeezed with the height-width ratio being altered? If such flexibility is allowed, it’d be great to visualize the performance vs. the degree of distortion.
- Line 249: “*During inference, its embedding for the graph structure can contain up to 10 hots, while during training, it is only up to 1-hot*”. Does the performance exhibit any degradation across inference examples from 1-hot up to 10-hot? If yes, how does the degradation look like versus #hots?
- Line 263: “*… viewed in 3D from a certain camera angle …*”. How is a 3D image input represented? Is it still a 2D matrix with 3 channels or is it represented as a 3D scene with depth?
- Line 263 again: I suppose the camera angle may distort a perpendicular relation so that it looks like an acute angle. And this distortion varies across camera angles. How did you address this and how did you choose camera angles when you constructed this dataset?
- Line 264: “*each test task consists of a tuple of three images*”. Why three images at a time?
- This doesn’t have to answered thoroughly but I’d love to see your thoughts: How do you expect your approach to generalize to non-90-degree angles, like the top angle in “A”, or relations about arcs?


**Limitations:**

Broader social impact and limitations are discussed in the supplementary.

Here I briefly point out a few limitations and addressing any of these would make the paper stronger.
- The datasets used for evaluation cover limited number of concepts.
- The search process for a mask indicating “objectness” is inefficient.
- The universality of the set of primitive concepts and relations isn’t justified.
- The current algorithm is unable to evolve with dynamic expansion and compression as the number of concepts to model increases.

**Strengths And Weaknesses:**

Strengths:
- This is a good move towards systematic generalization. The paper presents a way to train a recognition model on elementary patterns and perform inference on hierarchically composed patterns, given that the complicated pattern can be symbolically described (i.e. via a graph).
- The experiments are solid and suggest possible applicability to realistic use cases: 2D-to-3D domain adaptation, and generalization to real-world images.

Weaknesses
- Each grid-world dataset used for evaluation contains only 3 pre-defined concepts. There is still a long way to go in order to more rigorously test the model’s compositional generalizability.
- Object localization is achieved through a somewhat clueless search with location candidates sampled from a distribution. This might lead to an efficiency issue once the task difficulty, image resolution or object crowdedness is scaled up.
- At present, all primitive concepts and relations are hand-crafted, but the paper didn’t explain the universality of those primitives in a general object recognition task. For example, the current task setup only touches upon parallel vs. perpendicular, but it is unclear how to define a template for an “acute angle” since it is defined as a range. In other words, the question is: how much space of the visual territory do you expect that can be approximated by your set of primitives? In order to move towards recognizing real-world objects, what other primitives should be included or what bottlenecks might need to be conquered?
- It would be great to demonstrate how the pool of primitive recognition models could be dynamically expanded and compressed to allow for progressively increasing demands.

---

> ### Author Response · Authors · 2022-08-02
> **Official Response (continued 3)**
>
> > Line 263: “… viewed in 3D from a certain camera angle …”. How is a 3D image input represented? Is it still a 2D matrix with 3 channels or is it represented as a 3D scene with depth?
>
> The 3D image in Section 3.2 is represented as a 2D matrix with RGB channels, similar to how CLEVR is represented as a 2D image of a 3D scene.
>
> > Line 263 again: I suppose the camera angle may distort a perpendicular relation so that it looks like an acute angle. And this distortion varies across camera angles. How did you address this and how did you choose camera angles when you constructed this dataset?
>
> In constructing the dataset, we have fixed the camera angle. Different locations of the angle will make the perpendicular relation look like different acute angles in an image. This is completely fine, as explained above, in that as long as the dataset contains concept instances with such intrinsic variation, the learned EBM is able to recognize it. This is supported by the empirical result that the classification and detection accuracy for 3D images is well above “statistics” baseline.
>
> Line 264: “each test task consists of a tuple of three images”. Why three images at a time?
>
> This is because in this dataset, we have 3 compositional concepts, and for each concept we show one example. For a dataset with N compositional concept, we will then show N images where each image corresponds to one concept.
>
> > This doesn’t have to answered thoroughly but I’d love to see your thoughts: How do you expect your approach to generalize to non-90-degree angles, like the top angle in “A”, or relations about arcs?
>
> As explained above, our method can handle quite general relations, like non-90-degree angles or relations about arcs as long as a dataset demonstrating the intrinsic variation in these concepts is provided.
>
> **Summary**
>
> Above all, thank you for the detailed review and questions! We hope that we have resolved your concerns and answered your questions.

---

> ### Author Response · Authors · 2022-08-02
> **Official Response (continued 2)**
>
> **Below are answers to the questions:**
> > Line126: “*We want to determine if the concept c appears in a given image x. We need to marginalize over the mask m.*” How sensitive is your approach to the tightness of a mask? For example, does the concept c have to be fully occupying the entire masked space to be recognized?
>
> As is shown in Eq. (2), we use the mask $m_n^K$ at the last step K of SGLD to feed into the EBM: $E_{X,M,C}(x,\tilde{m}_n^K,c)$ and obtain the energy for concept c. We find empirically that the SGLD process already can find a quite good mask $\tilde{m}_n^K$ that locally minimizes the energy, where the mask overlaps quite well with the concept instance to be discovered. The more the mask differs from the mask for the concept, the higher the energy. We find that our model has reasonable tightness for the mask: not too tight to make the SGLD hard to optimize to find the correct mask, and not too loose to not be able to differentiate a correct mask and a perturbed version of the mask.
>
> > Line 149: “the mask M is the maximum of all the masks”. I suppose each constituent is being recognized independently, right? Then is it possible that the predicted masks don’t form a contiguous region, which would indicate a detection error? How would you plan to address this case?
>
> This question is about the Hierarchical Composition Rule (Eq. 3) that detects a compositional concept given a graph specification of constituent concept and relations. The different masks $m_i$, $m_{j1}$, $m_{j2}$ are independent variables that are optimized jointly by SGLD through a common compositional EBM energy landscape, and only the mask configuration that satisfies all concept and relation as specified will result in a low energy. It is entirely possible that the predicted masks don’t form a contiguous region, which is completely normal. For example, we can have a compositional concept like “parallel-line” that consists of two disconnected lines and a relation requiring that they are parallel. The constituent masks for this compositional concept are not contiguous, which is expected.
>
> > Line 230: “The hierarchical concepts to be classified and detected are three characters which we term Concept1, Concept2, Concept3”. These three concepts look like random patterns. How did you choose them in the first place?
>
> In this dataset, we have three primitive relations, “inside”, “enclose” and “non-overlap”, and two primitive concepts, “rectangle” and “E-shape”. The main goal of this dataset is to test whether our method can detect different relational graph structure given the same number of concept nodes. Concretely, given 2 “rectangle” concept instances and one “E-shape” instance, there are limited ways to form a compositional concept with different relational graph structures: Concept1 is where “E-shape” is not inside any of the “rectangle”. Concept2 is “E-shape” and “rectangle”_1 are both inside the other “rectangle”_2, and “E-shape” not inside “rectangle”_1. Concept3 is that “Eshape” is enclosed by one “rectangle” which is also enclosed by another “rectangle”. All three compositional concepts have the same number of constituent concepts but different relation graphs. We have added this discussion to the Appendix A.9 of the revised paper.
>
> > Does your approach exhibit any flexibility in recognizing concepts that have been slightly stretched or squeezed with the height-width ratio being altered? If such flexibility is allowed, it’d be great to visualize the performance vs. the degree of distortion.
>
> In the dataset provided to training the concept EBMs, we have found that it is beneficial to have concepts with varying height-width ratio. For all concept dataset we provide for training, e.g. “line”, “rectangle”, “E-shape”, we all sample varying height-width ratio, which helps the EBM recognize the intrinsic variation of a concept. This answer is also connected with the above response to the “acute-angle”, where it is beneficial to provide the training data that contain the intrinsic variation of the concept.
>
> > Line 249: “During inference, its embedding for the graph structure can contain up to 10 hots, while during training, it is only up to 1-hot”. Does the performance exhibit any degradation across inference examples from 1-hot up to 10-hot? If yes, how does the degradation look like versus #hots?
>
> In the current datasets, we don’t have compositional concepts with many different #hots. We will perform a more thorough study with more compositional concepts in future revised version.

---

> ### Author Response · Authors · 2022-08-02
> **Official Response (1)**
>
> We thank the reviewer for the positive review, and glad that the reviewer recognizes the strengths of our paper in the context of systematic generalization, and solid experiments with possible applicability to realistic use cases. Below we address the reviewer’s concerns and questions.
>
> **Under “Weaknesses”**:
> > Each grid-world dataset used for evaluation contains only 3 pre-defined concepts. There is still a long way to go in order to more rigorously test the model’s compositional generalizability.
>
> We agree with the reviewer that compositional generalizability is shown for 3 pre-defined compositional concepts for each dataset. However, even this capability is beyond what the current methods can do. Further extensions of our approach to more concepts are planned as future work.
>
> > Object localization is achieved through a somewhat clueless search with location candidates sampled from a distribution. This might lead to an efficiency issue once the task difficulty, image resolution or object crowdedness is scaled up.
>
> As stated in “detection” section in Section 2.2, detection with SGLD approximates a posterior sampling of $P_{M|X,C}(m|x,c)$ with a suitably learned concept EBM. When detecting compositional concepts, which are defined as a graph with constituent concepts as its nodes and constituent relations as edges, composing the corresponding EBM together actually limits the search space. Any configuration that violates at least one EBM will result in a high energy, which alleviates the complexity of the task when task difficulty, image resolution or object crowdedness scale up. Empirically, we find that our method performs well. (1) With many concept instances present: in detection task example Fig. 2 upper left, it can correctly select a compositional concept with 4 nodes (line concepts), 6 edges (relations) in the image that contains an underlying graph with 10 nodes (lines) and 45 possible edges. (2) Deal with more complex concepts and relations as shown by the results for HD-Concept dataset. (3) Scale to more realistic 3D images, both in learning concepts and relations of 3D images in Section 3.2, and also CLEVR experiments in Appendix A.12.
>
> > At present, all primitive concepts and relations are hand-crafted, but the paper didn’t explain the universality of those primitives in a general object recognition task. For example, the current task setup only touches upon parallel vs. perpendicular, but it is unclear how to define a template for an “acute angle” since it is defined as a range. In other words, the question is: how much space of the visual territory do you expect that can be approximated by your set of primitives? In order to move towards recognizing real-world objects, what other primitives should be included or what bottlenecks might need to be conquered?
>
> The EBMs in the ZeroC framework can learn general primitives as long as labeled data is provided to demonstrate the concept or the relation, even if that concept or relation is a range that contains some intrinsic variation. For example, to learn the “acute angle” relation between two lines (with varying angles), ZeroC only needs a dataset that contains many (image, mask1, mask2, “acute-angle”) tuples where the mask1 and mask2 identify the two lines in the image that form an acute angle, with different examples containing different angles. In other words, as long as the dataset contains enough data that identifies a concept/relation in a certain range, the ZeroC can learn such primitives. This is also shown in the HD-Concept dataset in Section 3.1, where “inside”, “outside”, “non-overlap” relation primitives are learned, and each relation has intrinsic variation. For example, the two masks for “inside” relation can have different positions, relative positions, and sizes.
>
> In order to move towards recognizing real-world objects, the ZeroC architecture is able to do that in principle. The main bottleneck is presented by labeled data, as we need to have detailed labels for many primitive concepts and relations that constitute real world objects. We will add the above discussion to the revised appendix of the paper.
>
>
> > It would be great to demonstrate how the pool of primitive recognition models could be dynamically expanded and compressed to allow for progressively increasing demands.
>
> Thanks for the suggestion! Our ZeroC architecture naturally supports continued expansion or compression of the EBM pool, as newly learned compositional concept EBMs can be dynamically added to the pool. Independently trained EBMs on new concepts/relations can also be added to the pool and composed together with existing EBMs. This is an exciting future direction, but is out-of-scope of the present paper, since here we focus on proposing the framework and demonstrate the zero-shot recognition and acquisition capability of ZeroC. We have added this discussion to the Appendix A.10 “Limitation of current work” section.

---

### Author Response · Authors · 2022-08-02
**General response**

We thank the reviewers for detailed and constructive reviews. We are glad that the reviewers generally recognize the novelty, significance, and soundness of our method. We provide responses to each reviewer, and hope that it will resolve the concerns. We have also updated the paper and Appendix in the revision, where the main updates are:

1. Added Appendix A.13 to explain the generality of our ZeroC approach, which is raised by reviewers maWL, AN9q, P8en. We also provide responses to each individual reviewer to this generality question. In summary, ZeroC is general both in terms of learning elementary concepts/relation as well as applicable to different datasets and scenarios (as shown in our experiments that we use the same model architecture for all 2D and 3D image experiments).

2. Added Appendix A.14 to explain the scalability of our ZeroC approach, which is raised by reviewer Yv6p, AN9q. In summary, our experiments in larger-scale datasets, including 2D to 3D domain adaptation in Sec. 3.2 and the CLEVR experiment in Appendix A.12 shows  applicability to more realistic use cases.

3. Added Appendix A.15 to explain the computational complexity of our ZeroC approach, which is raised by reviewer P8en. In summary, ZeroC's computational complexity remains fairly constant, measured in terms of single forward or SGLD step.

4. Expanded Appendix A.9 to explain the designing of HD-Concept dataset.

5. Expanded Appendix A.2 to explain the reason why we neglect the entropy term for training the EBMs.

---

### Meta-Review · Area_Chair_XBfS · 2022-08-31

**Recommendation:** Accept
**Confidence:** Certain

**Metareview:**

The focus of this work is on the introduction of a compositional reasoning model that enables zero-shot generalization. While there are a number of limitations (e.g. the small domain, limited concepts) but reviewers were content that the demonstrated results on low-resolution image domains proved the approach can scale to more realistic task complexity. The primary open challenge for richer tasks is the identification and training of elementary concepts -- a classification that may not hold.

**Award:**

No

---

### Decision · Program_Chairs · 2022-09-14

Accept